# Bias adjustment and downscaling of snow cover fraction projections from regional climate models using remote sensing for the European Alps

Michael Matiu[1], Florian Hanzer[2]

[1]Institute for Earth Observation, Eurac Research, Bolzano, 39100, Italy
[2]Department of Geography, University of Innsbruck, Innsbruck, 6020, Austria

*Correspondence to*: Michael Matiu (mmatiu.eurac@gmail.com)

**Abstract.**

Mountain seasonal snow cover is undergoing major changes due to global climate change. Assessments of future snow cover usually rely on physical based models, and often include post-processed meteorology. Alternatively, here we propose a direct statistical adjustment of snow cover fraction from regional climate models by using long-term remote sensing observations. We compared different bias adjustment routines (delta change, quantile mapping, and quantile delta mapping) and explored a downscaling based on historical observations for the Greater Alpine Region in Europe. All bias adjustment methods account for systematic biases, for example due to topographic smoothing, and reduce model spread in future projections. The trend-preserving methods delta change and quantile delta mapping were found more suitable for snow cover fraction than quantile mapping. Averaged over the study region and whole year, snow cover fraction decreases from 12.5 % in 2001-2020 to 10.4 (8.9, 11.5; model spread) % in 2071-2100 under RCP2.6, and to 6.4 (4.1, 7.8) % under RCP8.5 (bias adjusted estimates from quantile delta mapping). In addition, changes strongly depended on season and elevation. The comparison of the statistical downscaling to a high-resolution physical based model yields similar results for the elevation range covered by the climate models, but different elevation gradients of change above and below. Downscaling showed overall potential but requires further research. Since climate model and remote sensing observations are available globally, the proposed methods are potentially widely applicable, but are limited to snow cover fraction.

## 1 Introduction

Mountain regions store large amounts of precipitation in form of snow and ice, which provide essential water supply for downstream regions, affecting an estimated quarter of humanity (Immerzeel et al., 2020). Global warming resulted in significant changes of the cryosphere with melting glaciers and shifts in the timing and abundance of snow (Huss et al., 2017), which already affected the hydrological cycle (Morán-Tejeda et al., 2014) and will continue to do so in the future (Hanzer et al., 2018). These changes imply consequences on water supplies for domestic use, hydropower, and agriculture. Seasonal snow cover responds rapidly to climate variability and change, in contrast to glaciers, which are out of balance with

current climate and will, to some extent, continue to melt even if climate targets are achieved (Marzeion et al., 2018). Finally, besides acting as water storage, snow cover causes a significant atmospheric feedback due to its high albedo, modulating mountain weather (Wallace and Minder, 2021) and causing large uncertainties in climate projections of northern hemisphere land warming (Thackeray et al., 2018).

Snow cover can be modelled using a large variety of models, which can be roughly grouped into conceptual empirical models (e.g., temperature index models such as Hock, 2003), complex energy-balance models with snow physics (Brun et al., 1989), and simplified energy-balance models with few layers, which are used in land-surface schemes of climate and hydrological models (e.g., Zanotti et al., 2004). In order to estimate future snow cover, conceptual empirical models can fail because climate change violates the assumption of stationarity, while the most complex energy balance models might be computationally unfeasible or accumulate artefacts in long-term simulations.

Recently, regional climate models (RCMs) have become a feasible alternative to study large scale snow cover (Räisänen and Eklund, 2012), even in complex terrain such as the European Alps (Steger et al., 2013), owing to increases in resolution and modelling performance. RCMs dynamically downscale global GCMs (general circulation models) for a limited domain but with higher resolution. Using snow cover output directly from RCMs, instead of taking meteorological forcing from RCMs and feeding it into dedicated snow models, has some benefits. First, it provides a consistent physical signal with land-atmosphere feedbacks. Second, it removes the need to perform bias adjustment of meteorological input for the dedicated snow model. The main downside of RCMs is their coarse resolution and limited representation of snow processes, which can be a limiting factor especially for mountain areas. For example, the EURO-CORDEX (European branch of the Coordinated Regional Climate Downscaling Experiment) scenarios for Europe are available at 0.11° horizontal spacing. However, single higher resolution runs of RCMs at 1-5km are available (Warscher et al., 2019; Lüthi et al., 2019), but they still lack the breadth of the EURO-CORDEX ensemble with up to 55 members (Coppola et al., 2021), which allows to assess multiple scenarios and model uncertainty.

Additionally, RCMs suffer from biases, for instance in temperature and precipitation (Vautard et al., 2021), which would be the meteorological forcing for dedicated snow models, but also from biases caused by the relatively simple snow schemes of RCMs. In high mountain regions, evaluations of snow from RCMs are challenging because of a general lack of suitable reference data and scale mismatches between observations and models. The arguably most relevant snow parameter, snow water equivalent (SWE), is also the most difficult to estimate. In-situ observations are sparse, and estimates based on remote sensing suffer from large uncertainties (Largeron et al., 2020). For the European Alps, Terzago et al. (2017) evaluated SWE from EURO-CORDEX RCMs using an array of remote sensing and reanalysis products, and found a large spread in reference data sets, locally large overestimation of SWE, and differences between GCM and renalysis driven RCMs. Using an interpolated SWE data set based on in-situ data in Switzerland, Steger et al. (2013) found a general underestimation of SWE for elevations below 1000 m and overestimation above 1500 m. On the other hand, Matiu et al. (2020b) focused on different snow parameters, namely snow depth from in-situ observations and snow cover fraction from remote sensing, and found a good agreement between RCMs and observations, when accounting for elevation and temperature differences

between observations and models. It is likely that scale mismatches (low vs. high resolution grids or point vs. grid cell),
associated elevation biases, and the different reference data set uncertainties are causing these contradicting results.

Before climate model output can be used for climate change assessments or impact models, it usually undergoes some post-processing, such as bias adjustment and downscaling. These serve to overcome systematic biases between observations and model output, which can be caused by model inadequacies, inherited biases in RCMs from their driving GCMs, or biases associated to the mismatch between spatial resolution of reference observations and model. The reference observations can be points or grids, are often limited in extent compared to RCMs, and feature, in case of grids, typically higher resolutions.

The simplest form of bias adjustment is the delta-change (DC) approach, where the mean climate change signal (e.g., in temperature) is superimposed on the observation series. However, DC cannot reflect any change in the future distribution of the considered variable. The most widely used approach for bias adjustment is quantile mapping (QM), which can simultaneously perform downscaling, too. QM matches observed and modelled distributions and the non-parametric variant performs better in reproducing observed climatology than parametric versions (Gudmundsson et al., 2012). Since QM has been show to modify trends in a few cases (Maurer and Pierce, 2014), quantile delta mapping (QDM) was developed, which represents a trend-preserving QM approach (Cannon et al., 2015). The flexibility, performance, and ease-to-use has made QM or QDM a standard approach for national climate change assessments, see, for example, Switzerland (CH2018, 2018) or Germany (Krähenmann et al., 2021).

For assessing future changes in snow cover based on climate model scenarios, two methods are mainly employed. The first is to use downscaled and bias adjusted meteorological forcing from climate models to drive dedicated snow or hydrological models (DeBeer et al., 2021; Hanzer et al., 2018). The second is to use directly snow cover output from climate models (see above). However, the availability of long-term high-resolution satellite imagery has enabled a third option: To use remote sensing for bias adjustment and downscaling of RCM snow cover. To the best of our knowledge, this has not yet been performed. We restrict the study to snow cover fraction, which is, in contrast to snow depth and SWE, globally available on a high spatial resolution and with high accuracy. The presented method has therefore a global potential for application.

The aims of this study are to bias adjust and downscale snow cover fraction from RCMs using remote sensing observations for the European Alps, and to compare this, for a limited area, to the use of a dedicated snow model forced by downscaled RCM output. The motivation behind the statistical adjustment of snow cover fraction from RCMs is that the biases are systematic. They were shown to be mainly caused by orography and temperature, partly also precipitation, mismatches (Matiu et al., 2020b). These systematic biases are consistent across time, and thus future change estimates can be statistically adjusted. While bias adjustment cannot add information beyond what is contained in the RCM, it can reduce model spread. Additionally, it can make information on future projections more meaningful compared to solely providing change estimates, which are sometimes hard to interpret. Unbiased absolute values are better for climate change information and for impact assessments, which often depend on absolute thresholds, and for which biased estimates would not be representative. By exploiting the morphological dependence of snow cover on topography, downscaling can improve local spatial patterns of

RCM snow cover fraction. Finally, the comparison between downscaling RCMs and using a snow model shall highlight benefits and limitations of both approaches.

The study combines the proof-of-concept of applying bias adjustment and downscaling to snow cover fraction with its application to assess future scenarios of snow cover fraction over the European Alps. The remainder of the paper is structured as follows. Section 2 introduces the study region and data sets. Section 3 explains the methods used for bias adjustment and downscaling. Section 4 presents results and discussion, and Section 5 the conclusion.

## 2 Data

### 2.1 Study area

The study region (Fig. 1) encompasses the European Alps and approximately spans from 43 to 48.5° N and from 5 to 17° E, which roughly corresponds to the Greater Alpine Region (Auer et al., 2007). The large-scale climatic setting includes influences from the Atlantic Ocean, the Mediterranean Sea, and the European continent. The region is characterized by complex topography with strong elevational gradients. The comparison of statistical downscaling to a dedicated snow model is performed for a small subset, the Ötztal Alps region in Austria (1850 km$^2$, 862–3770 m a.s.l., Fig. 1b and 1d); see Hanzer et al. (2018) for a detailed description of the Ötztal Alps region.

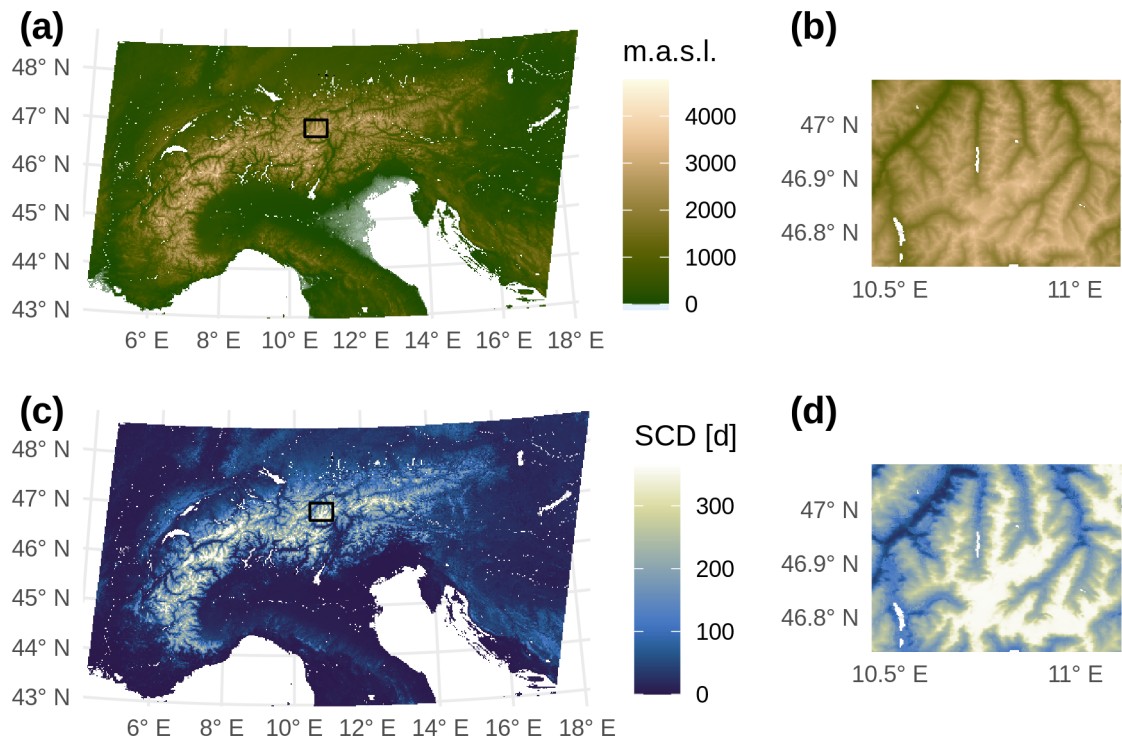

Figure 1: Topography (a-b) and average annual snow cover duration (c-d) of the study region, the European Alps, (a, c) as well as the Ötztal Alps region (b, d). The bias adjustment and downscaling has been performed on the whole area denoted in (a) and (c). Dedicated snow model (AMUNDSEN) simulations were available for the Ötztal Alps region (b, d), which is also indicated with a tiny square in (a) and (c). Snow cover duration maps are based on remote sensing and averaged over the hydrological years 2001-2020 (see also Sec. 2.2).

## 2.2 Observed snow cover fraction from remote sensing

As for remote sensing observations, we relied on MODIS (Moderate Resolution Imaging Spectroradiometer), because it offers the best tradeoff between temporal availability (two decades, daily) and spatial resolution (250 m) to perform a downscaling – in contrast to coarser products such as based on AVHRR (Advanced Very-High-Resolution Radiometer), which have a longer period into the past (starting in the 1980s), but are of coarse resolution and less quality for complex mountain terrain than higher resolution sensors. A cloud filtered product was used (Matiu et al., 2020a), which is based on the snow maps developed in Notarnicola et al. (2013a). The processing included a sequence of spatial and temporal filters to remove nearly all cloud coverage. More specifically, it included a mean filter to correct for errors in misclassifications of snow vs. clouds, which sometimes occurred at the edges of cloudy and snowy areas. This was followed by a conservative

temporal filter, which is based on the persistence of snow, and which filled short gaps between periods of snow or absence of snow. Then an elevational filter was applied that filled cloud pixels above a snow line and below a land line with the respective classes. Finally, a greedy temporal filter was applied, which filled values with the next available observation in time. This was often achieved within three to seven days, but if the next available observations was more than ten days away, the pixel remained cloud. For more specific details, we refer to Matiu et al. (2020a), where an additional step is described, namely the merging of Terra and Aqua acquisitions. Here, we only used Terra, in order to extend the temporal extent to 2000.

Consequently, nearly cloud-free binary (snow/land) snow cover maps were available at daily scale for the complete period 2000-02-24 to 2020-08-23 at 250 m resolution in Lambert azimuthal equal-area projection for the domain denoted in Fig. 1. While the actual horizontal resolution of the maps is 232 m, the approximation 250 m will be used throughout the manuscript for simplicity. Nearly cloud-free means that less than 0.1% of observations (over all pixels and all days) contained clouds. These cloudy pixels were removed from the subsequent analysis.

The high-resolution binary snow maps were aggregated into low-resolution snow cover fraction maps that match the RCM resolution of 0.11°, which is approximately 8.6 by 12.2 km for the Alps. Each low-resolution grid cell then contained 1961 (37*53) high-resolution pixels. From now on, the term pixel shall refer to the high-resolution area (250 m by 250 m) and grid cell to the coarse-resolution area (0.11° by 0.11°), for both MODIS and RCMs.

To derive annual snow cover duration (SCD) maps, we used hydrological years defined such as to maximize the available data within the MODIS period. The split was in summer, which is anyway the least important period for seasonal snow in mountains. A hydrological year is defined here as starting August 1 and ending July 31, and designated by the year it ends. The past SCD climatology (Fig. 1) is thus based on the (hydrological) years 2001 to 2020, which covers the period 2000-08-01 to 2020-07-31. For simplicity, the term year will be used as substitute for hydrological year from now on, thus also when referring to the climate model data.

### 2.2.1 Scale issues in the study area

The aggregation of maps of snow cover duration from 250 m pixels to 0.11° grid cells creates scale issues that hinder comparisons between high and low resolution. The aggregation smoothens the spatial patterns and creates systematic differences between high and low elevations (Fig. S1). When the SCD maps are then further aggregated by elevation, the resulting SCD differs substantially between high and low resolutions, especially between 1000 and 2000 m (Fig. S2), and despite the fact that the distribution of pixel/grid cell elevations is almost identical between high and low resolutions (Fig. S2b). At 1500 m, SCD from the low resolution map is more than 15 d (i.e., approximately 18%) higher than in the high resolution map. Consequently, also future estimates of SCD cannot be compared between high and low resolutions without introducing the same errors from scale issues. This holds for absolute numbers of SCD; however, it's still possible to compare future change estimates, since subtracting past from future also subtracts the biases introduced by scale mismatches.

## 2.3 Snow cover fraction from regional climate models

The EURO-CORDEX ensemble consists of 11 RCMs driven by 8 GCMs from CMIP5 (Coupled Model Intercomparison Project Phase 5); see Coppola et al. (2021) for more information on the general ensemble setup. However, not all models provide all variables and/or all emission scenarios. For instance, temperature and precipitation are available from all models, but snow parameters such as SWE or snow cover fraction are only available for a subset of models. Regarding scenarios, we used the RCP2.6 (representative concentration pathway) and RCP8.5 scenarios, where RCP2.6 is likely to keep global warming below 2° C until 2100, while RCP8.5 corresponds to approximately 4 to 5° C global warming. Regarding snow cover fraction (SNC), the available ensemble for this study included, for RCP8.5, 6 RCMs driven by 6 GCMs with a total of 29 simulations, and, for RCP2.6, 4 RCMs driven by 5 GCMs with a total of 8 simulations (Table S1). The list of used RCMs is CLMcom-CCLM4-8-17, CLMcom-ETH-COSMO-crCLIM-v1-1, CNRM-ALADIN63, IPSL-WRF381P, KNMI-RACMO22E, and SMHI-RCA4. Even though DMI-HIRHAM5 also provides SNC, we excluded it because the SNC values over the Alps were unrealistically low, although snow depths were well reproduced (Matiu et al., 2020b).

The RCM SNC maps were reprojected onto the low-resolution MODIS maps using nearest neighbor resampling in order to have a one-to-one correspondence of grid cells in the spatial domain. Nearest neighbor was favored over other resampling methods, such as bilinear, because it preserves the two-sided bounded nature of SNC, which goes from 0 to 1, and thus keeps the same limits, while bilinear resampling can introduce lower maxima or higher minima.

Some models display snow accumulation issues (Terzago et al., 2017; Matiu et al., 2020b; EURO-CORDEX Errata, 2021) and affected grid cells were removed based on thresholds on snow water equivalent or snow depth, as described in Matiu (2020b). These were mostly grid cells with the highest elevation and the number of affected cells was between 0 and 233, depending on model, out of a total of approximately 5000 land grid cells in the study domain (see Fig. S3 and Table S2 for location and number of affected cells by RCM). Spatial averages and ensemble means are based on the common subset of grid cells available to all models. KNMI-RACMO22E was strongly affected by snow accumulation, and thus we removed it from results that show spatial/ensemble means, since otherwise too many grid cells would have to be removed. Consequently, for calculating ensemble means, 23 simulations were available for RCP8.5, but only 4 for RCP2.6. This imbalance between future scenarios was unfortunately unavoidable. While it would have been possible to restrict the number of simulations to the same GCM-RCM pairs for both RCP2.6 and RCP8.5, we still decided to take all possible simulations in order to have a better estimate of the model spread. However, model spread is likely underestimated for RCP2.6 due to the low number of available simulations.

## 2.4 Snow cover fraction from AMUNDSEN

The fully distributed snow and hydroclimatological model AMUNDSEN (Strasser, 2008), now available as openAMUNDSEN in Python (Warscher et al., 2021), has been previously applied to study the future snow and ice evolution

in the Ötztal Alps region in Austria (Hanzer et al., 2018); see Fig. 1b and 1d for the area. AMUNDSEN dynamically resolves the mass and energy balance of snow and ice and has been driven by future meteorology based on EURO-CORDEX RCMs, which includes a subset of the same RCMs mentioned above. The input meteorology has been bias adjusted and downscaled using QM to point scale, and further temporally disaggregated and spatially distributed to provide 3 hourly forcing at 100 m horizontal resolution for the whole catchment. For more details, see Hanzer et al. (2018). The modelled snow water

equivalents were converted into binary snow/land indicators using a threshold of 5 mm. We also evaluated a threshold of 15 mm but found differences to be negligible. Snow cover fraction was then calculated by averaging over time (e.g., months), space (which includes elevation bands), or both.

For the comparison, we decided to focus on ensemble means, and not compare results between individual GCM-RCM pairs directly. Only few GCM-RCM pairs overlap between Hanzer et al. (2018) and this study. In addition, the QM in Hanzer et

al. (2018) has been applied using the period 1970 to 2005 as baseline, while the baseline in this study was 2001 to 2020. Finally, the ensemble size is similar, since Hanzer et al. (2018) used 14 GCM-RCM for RCP8.5 and 3 GCM-RCM pairs for RCP2.6, compared to 19 and 3 here, respectively (CNRM-ALADIN63 was removed only from the comparison to AMUNDSEN, because half of the Ötztal Alps area was affected by snow accumulation).

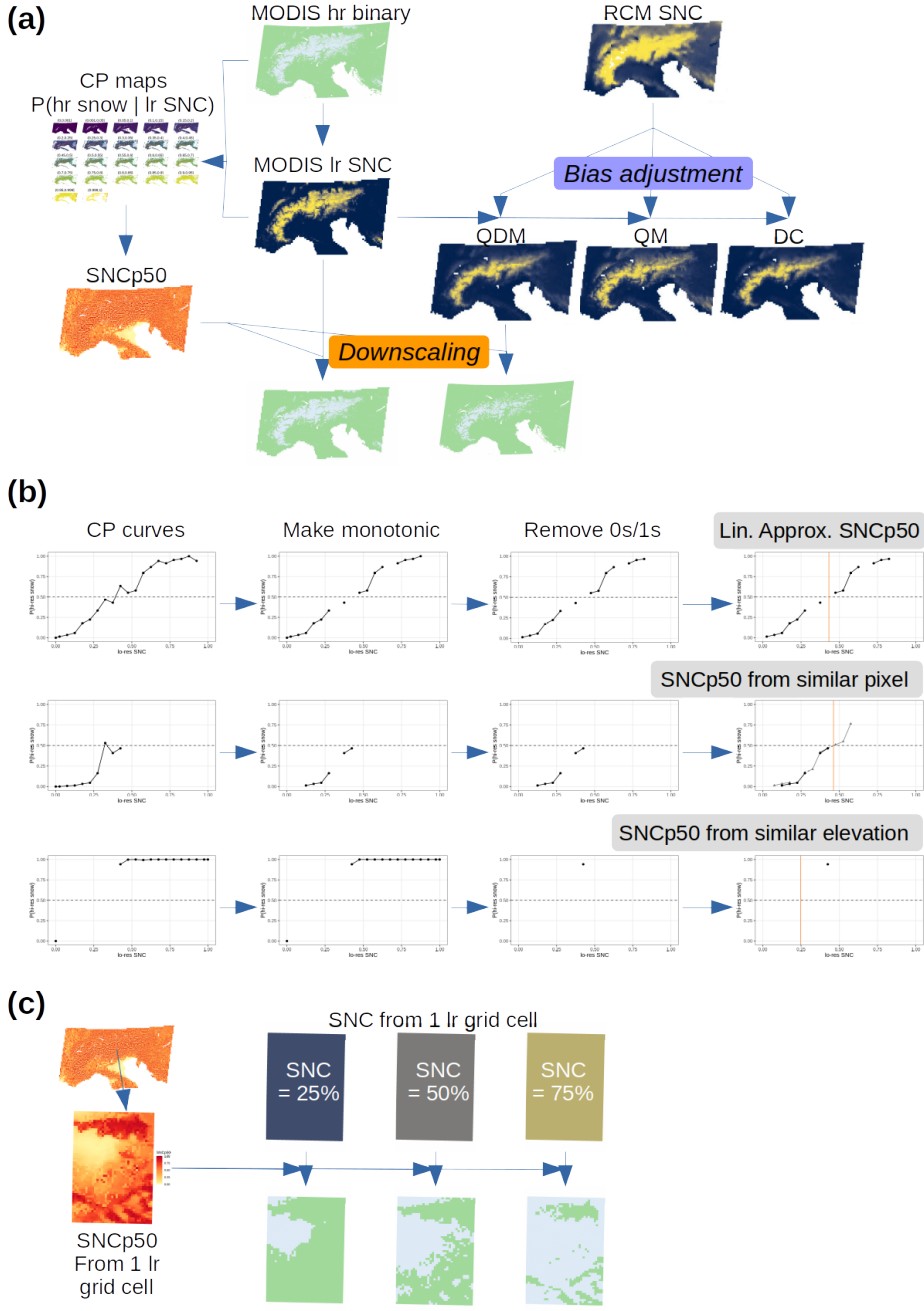

**Figure 2. (a) Overview of methodology. (b) Detailed view of the estimation of SNCp50 based on CP curves. CP curves show the probability of the respective pixel being snow covered as a function of the encompassing grid cell SNC. (c) Downscaling exemplified at one low-resolution (lr) grid cell. The SNCp50 values determine the conversion from lr SNC to hr binary snow, which is shown for three example SNC values. Abbreviations: lr (low-resolution), hr (high-resolution), SNC (snow cover fraction), RCM (regional climate model).**

## 3 Methods

The overall methodology is summarized in Figure 2a. It consists of two separate steps, bias adjustment and downscaling, which are both explained in detail below. MODIS observations are used overarchingly: as reference climatology, for bias adjustment, to derive the downscaling relationship, and to validate the downscaling approach.

### 3.1 Bias adjustment of snow cover fraction

We compared four different bias adjustment methods, which are routinely applied for temperature and precipitation series, in their applicability for snow cover fraction: DC, QM (Gudmundsson et al., 2012), QDM (Cannon et al., 2015), and multivariate QDM (Cannon, 2018). In all cases, the past refers to years with MODIS observations available, that is 2001 to 2020.

For the climate model runs, the historical period, which goes from 1950/70 to 2005, was merged with the RCP scenario run, which covers 2006 to 2100, in order to have the same common period for the past as available from MODIS (2001 to 2020). Thus, for applying the bias correction, each scenario had its own past time series (i.e., for each RCP scenario), while usually the calibration is performed on the same historical run for all scenarios. However, this was not possible here, since the overlap between historical period and MODIS is only 5 years, which is too little to derive robust distributional estimates of the snow cover climatology. As future period, we considered 2071 to 2100. The bias adjustment was applied at the same spatial (0.11°) and temporal scale (daily).

For the DC approach, we calculated the multiplicative change ratios between the past and future from RCMs and applied it to the observations from MODIS. This was done separately for each grid cell and each month. For QM and QDM, we employed the standard routines with empirically derived distribution functions (Gudmundsson et al., 2012; Cannon et al., 2015), as available in the R packages qmap and MBCn, again month by month and grid cell by grid cell. For QDM, multiplicative change ratios were used. The multivariate QDM was applied in the spatial domain, thus the multivariate component was to account for the spatial correlation in SNC between grid cells. However, we found results to be almost identical to standard univariate QDM, and we do not show it further in results. We assume this similarity to be caused by the high spatial correlation in SNC, and the fact that this spatial correlation is similar in both model and observed series.

Because SNC is bounded not only at the minimum 0 but also at the maximum of 1, the standard QM and QDM algorithms were both modified as follows. The trace condition, which sets all values below a threshold (here: 0.001, also called trace value) to exact zeros, has also been applied to the maximum, so that all values above 0.999 were set to exact ones. In addition, the distribution of RCM SNC contained many exact zeros and ones in comparison to observed SNC, which was more regularly distributed across the [0, 1] interval, caused by the sub-grid variability in observations. This caused problems in estimating and matching the modelled and observed quantiles. To alleviate this issue, we added a random component to all SNC values near 0 and 1, where near means half of the trace value. The random values were randomly sampled from a uniform distribution with minimum the machine epsilon (the lowest value without rounding issue in floating point

arithmetic) and maximum half the trace value, i.e., effectively from the [0, 0.0005] interval. This random component helps in matching quantiles (and thus distributions) but breaks the temporal consistency in the bias adjusted SNC time series. Since it's applied in a distributional manner over all days in each month for a 20/30 year period, the monthly climatologies are

250 fine. But at the daily scale, the random component might lead to inconsistencies, such as sudden jumps in the snow cover fraction time series or increasing snow cover fraction in the melt season. Consequently, no estimates of interannual variability can be calculated.

### 3.2 Downscaling of snow cover fraction to binary snow

The proposed downscaling approach converts low-resolution snow cover fraction (SNC) from RCMs into high-resolution binary snow cover (snow/land), from which we extracted monthly and annual snow cover duration (SCD). The downscaling is based on the morphological dependence of snow cover (Premier et al., 2021). It uses a conditional probability (CP)

approach (Dong and Menzel, 2016) to define the relationship between snow cover fraction of a low-resolution grid cell and the probability of a high-resolution pixel being snow-covered or snow-free. The procedure first estimates these CP maps/curve, which are then used to derive the SNCp50 threshold (Figure 2a). SNCp50 is the SNC value, for which the probability of a pixel being snow covered is higher than 0.5. These SNCp50 values then convert continuous SNC from a grid cell into high-resolution binary snow pixels (Figure 2c).

We used the 20 years of daily MODIS snow maps to calculate the CP that a high-resolution pixel is snow-covered depending on the SNC of the low-resolution MODIS grid cell (which itself has been aggregated from the high-resolution maps). For this, we split the maps by grid cells. For each grid cell, the 20 years of daily SNC observations were divided into 22 bins with breaks 0, 0.001, 0.05, 0.1, … 0.95, 0.999, 1. These are SNC bins of width 0.05 with additional bins at the minimum and maximum to catch nearly exact zeros and ones. For each bin, we defined the CP of each high-resolution pixel as the fraction

of days each pixel was snow covered divided by the total number of days in the respective bin. Bins that contained less than 30 days were omitted, which were mostly with low SNC at high elevations and high SNC at low elevations (see Figure S4). For each pixel, this results in up to 22 empirically estimated probabilities that a pixel is snow-covered based on the encompassing SNC value (derived as the average of the range of the SNC bins). These correspond to the 22 CP maps (all pixels, one bin; e.g, Figure 2a or Figure S4) or CP curves (one pixel, all bins; e.g., Figure 2b or Figure S5d). The CP curve

with up to 22 points can be considered an empirical approximation to a smooth function that gives the probability of a pixel being snow-covered as a function of the encompassing grid cell SNC.

From these CP estimates, SNCp50 was derived in three ways. First, with a linear approximation (64 % of cases); if this failed, then using a similar pixel approach (7 % of cases); if this also failed, then with a similar elevation approach (remaining 29 % of cases). These steps are described in detail below.

SNCp50 is the x-value (SNC) at which the CP curve crosses y=0.5. This value can be extracted from the empirical CP curves via linear approximation. A sufficient condition for a unique solution is a non-strictly monotonic relationship between low-resolution SNC and probability of high-resolution snow, which is a physically valid assumption for any given high-resolution pixel. A non-monotonic curve could imply that the y=0.5 lines is crossed multiple times, thus resulting in multiple SNCp50 values. But because of noise and errors in the MODIS time series, monotonicity was not always the case, so we

selected the longest non-strictly increasing subsequence. Additionally, to have robust estimates of this SNCp50 threshold, we removed points with probability exactly zero and one, thus requiring some points that identify the curve (Figure 2b last row). The linear approximation worked for 64% of pixels.

In the remaining 36%, the linear approximation failed to estimate SNCp50, because either no empirical estimates were available (except for ones and zeros) or all were above/below 0.5 (see, e.g., point (3) in Figure S5d). For these pixels,

SNCp50 was imputed in two steps, first using a similar pixel approach and if this failed, with a simpler elevational filter.

For the similar pixel approach (Li et al., 2020), we selected another reference pixel with available SNCp50 that is similar with respect to, first, the sub-grid topography of the encompassing low-resolution grid cell and, second, to the high-resolution probability curves. For the first, similarity between low-resolution grid cells was assessed with the Wasserstein distance (also called earth-mover distance) using the high-resolution pixel elevations. While it would be possible to match

pixels of two grid cells (by, for example, taking the topleft pixels of both grid cells, then the next pixels to the right, etc. up to the bottomright pixel) and then calculating the Euclidean distance, this would not be a reasonable approximation of the difference in sub-grid topography between the two grid cells. If, hypothetically, a grid cell were flipped horizontally or vertically, its sub-grid topography variability would be identical, but the Euclidean distance from above would be different. The Wasserstein distance, on the other hand, would be zero for a flipped grid cell, since it is based on distributions.

Consequently, it is a suitable metric to estimate the similarity of the variability of pixel elevations between two grid cells. See Figure S6 for example elevation distributions and Wasserstein distances.

For each high-resolution pixel with missing SNCp50, we selected the 50 nearest low-resolution grid cells (including the low-resolution grid cell with missing high-resolution SNCp50); nearest in terms of the Wasserstein distance. Then, we calculated the mean absolute error (MAE) between CP curves for pixels that deviate at most 150 m of the missing pixel elevation and

have at least 5 values to compare CP curves. The SNCp50 from the pixel with minimal MAE was used to fill the gap. The similar pixel approach filled an additional 7% of SNCp50 values.

The remaining 29% of missing SNCp50 were mostly located above 3000 or below 500 m. For these, the second imputation step involved a simpler elevation filter, and no comparison of probability curves. Again, we selected the 50 most similar low-resolution grid cells in terms of the Wasserstein distance. All high-resolution pixels in these 50 grid cells were ordered

by their elevation difference to the gap pixel and up to 100 pixels with at most 150 m elevation difference were selected. The average SNCp50 from these up to 100 pixels was then used to fill the gap. After this step, <0.001% pixels were missing and these were omitted from the rest of the analyses.

We excluded glacierized pixels with more 10% glacierized area from the further analysis of the downscaling, because of a systematic bias (see validation in Sec. 4.4) in addition to the difficulties of distinguishing snow and ice with MODIS (Fugazza et al., 2021). In addition, they had a strong overlap with the already removed low-resolution RCM grid cells with snow accumulation (Sec. 2.3). Glacier extents were extracted from the Randolph Glacier Inventory 6.0 (RGI Consortium, 2017).

An example of the SNCp50 values is shown in Figure S5, which shows the expected negative relationship between SNCp50 and elevation, which implies that, as SNC increases from 0 to 1, the snow cover is more likely to be found going from high to low elevations. But while elevation is the main influence, SNCp50 can vary considerably for similar elevations (e.g., points (1), (3), and (4) in Figure S5 are approximately the same elevation) due to local terrain factors.

For the validation of the downscaling, we applied the procedure to the upscaled MODIS snow cover fraction maps and compared the downscaled maps with the original maps, which have been used in the upscaling, too. This comparison involves a contingency table for a binary classification (snow/land), where we define snow as positive outcome. From the numbers of true positives (TP, correctly downscaled snow), false positives (FP, downscaled snow, but actually land), true negatives (TN, correctly downscaled land), and false negatives (FN, downscaled land, but actually snow) we calculated the following metrics: Accuracy, which is the overall fraction of correct values (TP+TN)/(TP+FP+TN+FN), positive predictive value (PPV), which is the fraction of correctly downscaled snow of all snow TP/(TP+FP), and the negative predictive value (NPV), which is the fraction of correctly downscaled land of all land TN/(TN+FN). The PPV and NPV are similar to the sensitivity and specificity metrics, but adjust for the prevalence of each category.

The downscaling was applied for QDM bias adjusted low-resolution snow cover fraction only, and not for the other bias adjustment methods. QM showed artificial modification of trends (see Sec. 4.1), and DC is theoretically inferior to QDM, since DC only adjusts the mean, while QDM adjusts the whole distribution.

## 4 Results and Discussion

### 4.1 Bias adjustment and future changes in snow cover fraction

The RCMs reproduced overall seasonal and large-scale spatial patterns of past snow cover fraction well (cf. RAW in Fig. 3 and 4). For instance, Winter (December-February) snow cover fraction spatial patterns agreed not only for the high-elevation Alpine region, but also for lower elevation mountains, such as the northern Alpine foreland, Dinarides, or the Northern Apennines. However, because of the coarse resolution and smoothed model orography, the RCMs did not capture the fine scale complex patterns found in the Alps (Fig. 4). Additionally, the monthly areal averages of snow cover fraction were

over- and underestimated, depending on both RCM and GCM (Fig. 3), as has also been shown previously (Terzago et al., 2017; Matiu et al., 2020b). This model bias depended stronger on the RCM and only secondly on the driving GCM.

Applying DC, QM and QDM bias adjustment to past RCM output enforced it to match the distribution of observed SNC and consequently also reduced the model spread for the future (Fig. 3). In addition, it introduced the fine-scale spatial patterns into the smoothed model output (Fig. 4). QM and QDM, by definition, resulted in the same patterns for the past. Bias adjusted future estimates were similar for the two trend preserving approaches DC and QDM, which themselves differed substantially from QM. For example, QM showed less reduction in SNC under the RCP8.5 scenario for spring (March-May)

than DC and QDM (Fig. 3 and 4).

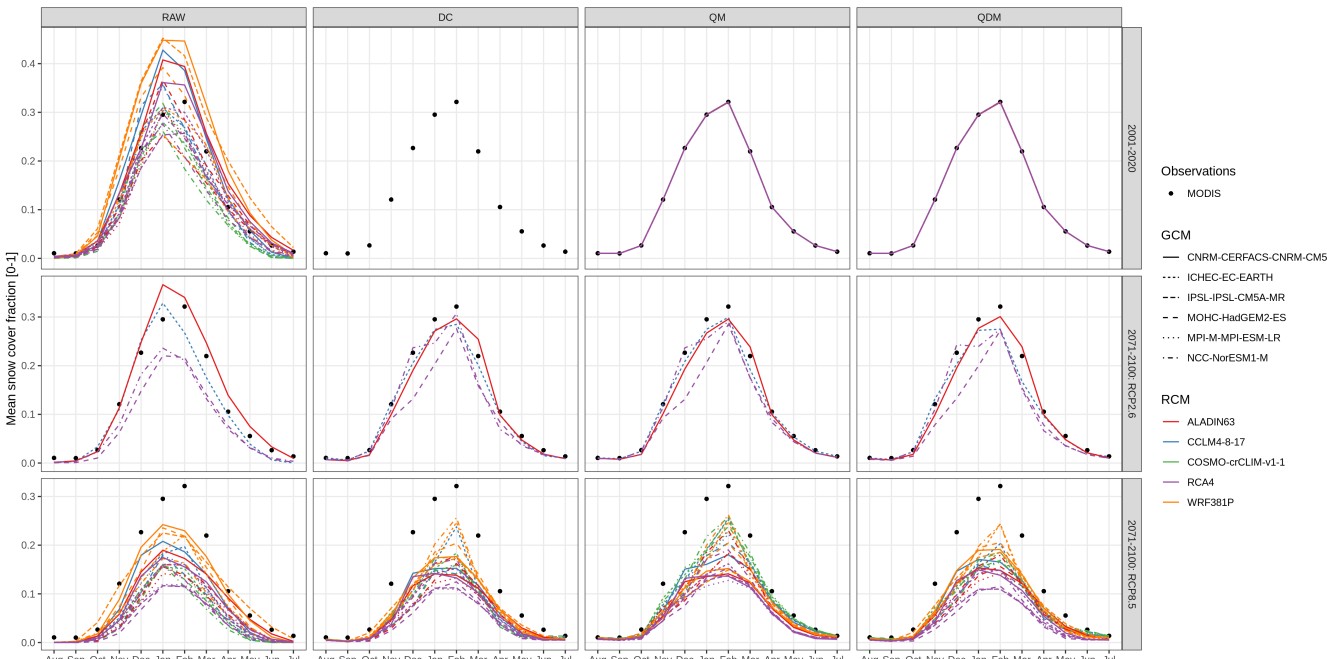

**Figure 3. Average monthly snow cover fraction over the whole study domain before and after bias adjustment. Black points denote observations from remote sensing for the period 2001-2020 (the same in all panels), and colored lines the regional climate model**
**(RCM) simulations with associated general circulation model (GCM). First row shows monthly averages for the past (2001-2020), while the middle and last row are for 2071-2100 averages for two emission scenarios (RCP, representative concentration pathway). Column RAW is for original RCM output, DC is the delta change approach, QM is quantile mapping, and QDM quantile delta mapping. Panel for DC and 2001-2020 shows no lines, since DC has no past RCM observations of its own.**

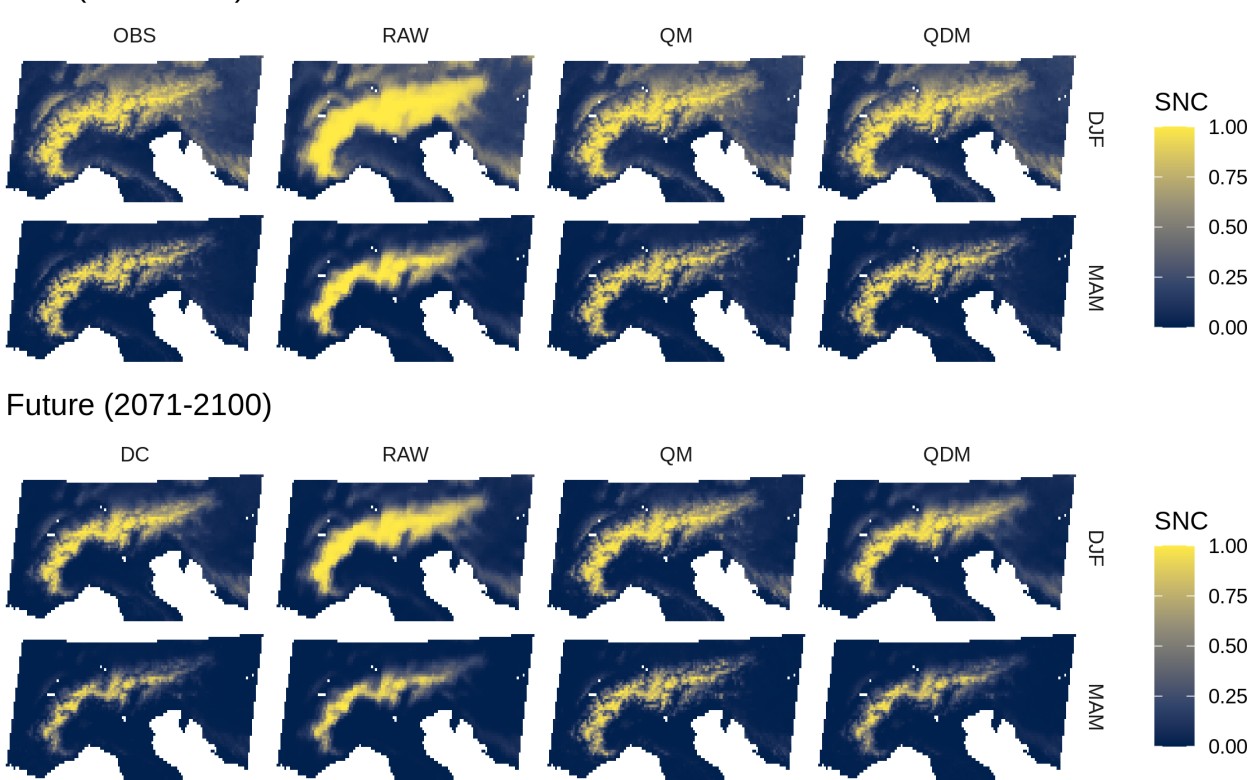

**Figure 4. Average seasonal snow cover fraction (SNC), as observed from remote sensing (OBS) and simulated with the CLMcom-CCLM4-8-17 regional climate model driven by CNRM-CERFACS-CNRM-CM5 under the RCP8.5 emission scenario. Abbreviations: remotely sensed observations (OBS), raw climate model output (RAW), quantile mapping (QM), quantile delta mapping (QDM), delta change approach (DC), December-January-February (DJF), March-April-May (MAM). For maps of the other climate models and emission scenarios, see (Matiu, 2021).**

Average winter SNC over the whole study domain was 29.3 % for the past (2001-2020) from MODIS observations, and the raw model mean was 30.2 (23.4, 43.3; model spread) %. For 2071-2100 under the low emission scenario RCP2.6, SNC decreased by 4.1 (2.4, 8.1) percentage points (pp) based on QDM, which corresponds to a relative reduction of 14.0 %. Under the high emission scenario RCP8.5 the reduction was 14.2 (10.1, 19.1) pp, which corresponds to 48.5 %. Observed

past spring SNC was 13.5 %, while the raw model mean was 13.0 (7.7, 21.7) %. Future changes under RCP2.6 were -2.7 (-4.5, 0.1) pp, in relative terms 20.8 %, while under RCP8.5 changes were -6.5 (-9.2, -4.6) pp, in relative terms 50.0 %. The estimates for RCP2.6 are based on a much smaller ensemble of only 4 GCM-RCM combinations, compared to 23 for RCP8.5, and thus are less likely to represent model uncertainty well.

Projected changes until the end of the century depended strongly on elevation, and the strongest absolute reductions in

winter SNC were observed between 400 and 2000 m and in spring above 1000 m (Figures S7 and S8, Tables S3 and S4). On

the other hand, relative reductions were strongest at the lowest elevations, and became gradually less with increasing elevation (Figure S9, Tables S5 and S6). Under RCP2.6, winter SNC decreased approximately 7 pp (15 %) at 1000 m elevation , while above 2000 m it's less than 2 pp (< 2 %) and the model uncertainty includes no change (Tables S3 and S5). Under RCP8.5, winter SNC decreases more than 15 pp between 400 and 2000 m elevation (corresponding to -21.1 % at 400

380 m and -3.4% at 2000 m), with strongest absolute changes at 1200 to 1400 m, which amount to -25.1 (-35.0, -17.7) pp or -12.0 (-27.3, -4.5) %. In spring under RCP2.6, strongest absolute reductions in SNC were observed at 1400 to 2000 m with more than 10 pp decreases in SNC (16 to 23 %). On the other hand, under RCP8.5, reductions in SNC were almost twice as large and remained high also above 2000 m compared to RCP2.6, where they gradually diminished, for example, at 2 200 to 2400 m, changes were -23.3 (-43.2, -4.7) pp or -27.3 (-50.9, -5.5) % under RCP8.5 and -6.0 (-12.1, -1.2) pp or -7.1 (-14.2, -

385 1.4) % under RCP2.6 (Tables S4 and S6). In addition, a high model uncertainty in projected changes of spring SNC above 2000 m under RCP8.5 was observed: The model spread ranged from almost no change to an approximate halving of SNC (Figures S7 and S9).

This model spread in spring SNC under RCP8.5 is likely caused by the snow schemes in the climate model's land surface schemes in combination with the projected temperature and precipitation changes, which directly affect SWE, and, since

SNC is parametrized on SWE, also SNC. Higher uncertainties are expected in spring because potential errors accumulate over the snow season. However, a detailed discussion on snow model processes and uncertainties is beyond the scope of this study, and better addressed in dedicated projects, such as ESM-SnowMIP (Krinner et al., 2018).

An in-depth view of the bias adjustment results at a single grid cell highlights the main differences between raw climate model SNC and observations as well as between QM and QDM (Figure S10). RCMs have more saturated SNC at both 0

(snow free) and 1 (snow covered) and thus display more often fully snow free or snow-covered conditions over time as compared to MODIS. This is likely caused by sub-grid variability, which is prominent in MODIS, since it's based on 250 m information. The trend-preserving attribute of QDM keeps the distribution of SNC identical between past and future when raw model SNC does not change, e.g., for the fraction of time, where the grid cell is fully snow covered (Figure S10). In the same situation, QM shows reductions in SNC. Additionally, QM has spurious breaks caused by applying the method month-

by-month, but not QDM. While these breaks could be alleviated by applying the bias adjustment with a moving window approach (e.g., 3-6 months) or using the whole year, QM still suffers from artificial modification of trends, as has been partly shown before for precipitation  (Maurer and Pierce, 2014; Maraun, 2013). Consequently, it should be treated with caution for snow cover fraction, too. For the downscaling below, we thus only used results from QDM.

## 4.2 Downscaled projections of snow cover duration

Downscaled projections of high-resolution snow cover duration (SCD) based on low-resolution snow cover fraction (SNC) showed decreases in annual SCD under both emission scenarios, but much stronger under the high emission RCP8.5 scenarios (Figure 5). Similar to Section 4.1, changes strongly depended on elevation (Figure S11, Table S7). In absolute terms, SCD decreased stronger with increasing elevation, while in relative terms, the reductions were highest at the lowest elevations (Table S8). For example, averaged over all pixels between 800 and 1000 m in the study area, SCD decreased by 9 d (3, 17) under RCP2.6 and by 26 d (19, 33) under RCP8.5. At higher elevations of 1800 to 2000 m, SCD decreased by 25 d (11, 47) under RCP2.6 and by 68 d (41, 106) under RCP8.5, while at 2800 to 3000 m SCD decreased by 35 d (21, 56) under RCP2.6 and by 92 d (49, 163) under RCP8.5. In relative terms, these reductions amount to 22 %, 15 %, and 11 % under RCP2.6 for 800-1000 m, 1800-2000m, and 2800-3000m, respectively, and 64 %, 41 %, and 30 % under RCP8.5.

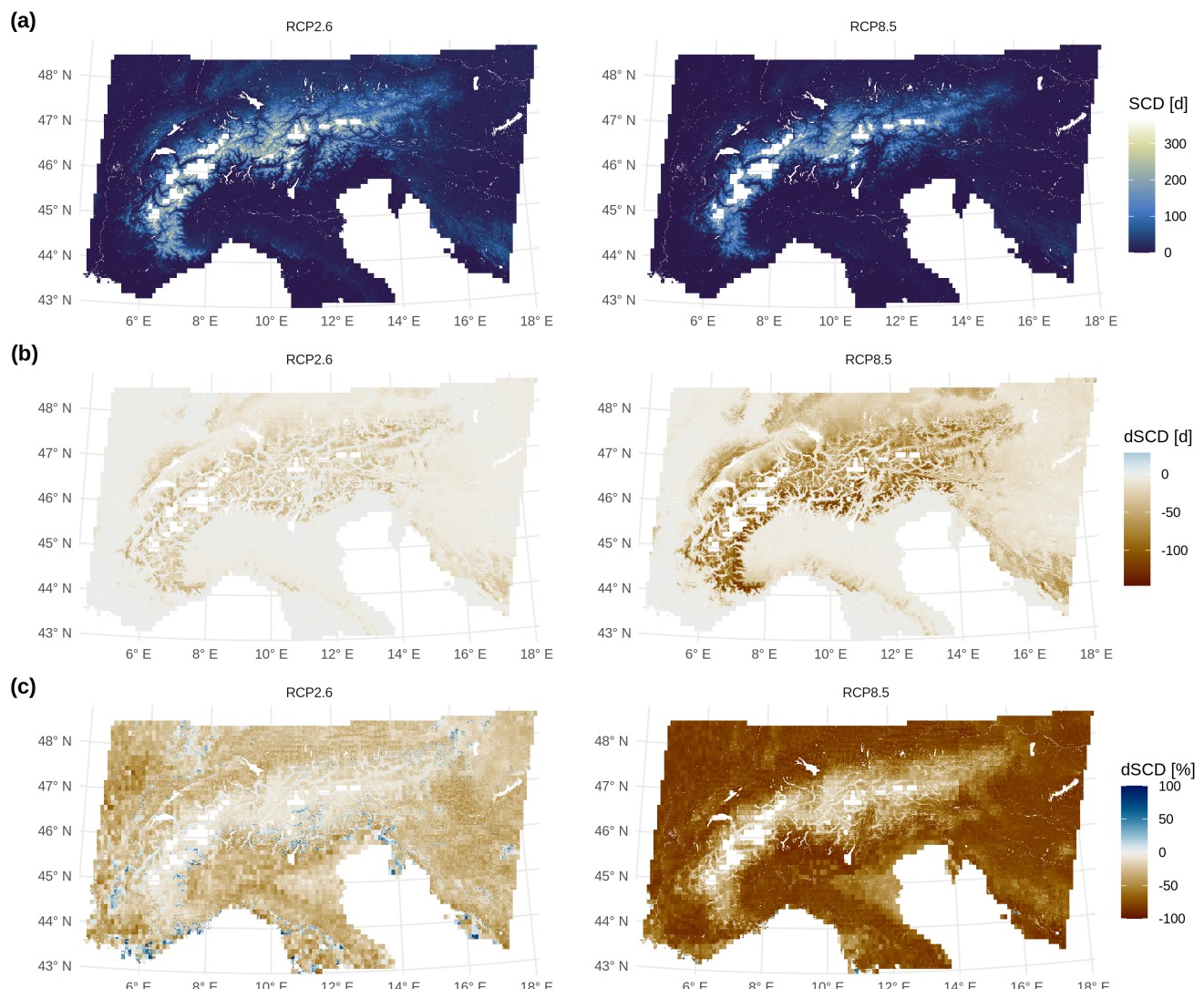

**Figure 5.** Future 2071-2100 annual snow cover duration (SCD) maps and differences to past (dSCD). (a) Downscaled SCD maps for low and high emission scenarios (RCP, representative concentration pathway) based on an ensemble of 4 models for RCP2.6 (regional climate models driven by general circulation models) and 23 models for RCP8.5. Empty areas denote pixels removed because of snow accumulation issues (see methods), glaciers, or water bodies. (b) Differences between future (2071-2100) and past (2001-2020) model output in absolute days. (c) Relative differences; pixels with > +100% difference omitted, because low values of observed SCD caused noise in relative estimates for low SCD (most of the remaining positive changes are for areas with SCD <= 5 d, too).

The European Alps have a prominent north-south climatic divide (Auer et al., 2007), which manifests itself in snow cover duration, too. Taking anomalies of SCD by elevation shows, on average, higher SCD north of the main ridge and lower SCD south (Figure S12a). These patterns were reproduced in RCMs, too, and changed in the future period: Comparing RCP2.6 to

RCP8.5, the north-south gradient in SCD was less strong for lower elevations and more pronounced for higher elevations (Figure S12). In addition, a stronger relative decline in SCD was observed south and west of the Alps compared to north and east (Figure 5c) under RCP2.6. An analysis of station snow depth and SCD trends over the last five decades in the Alps similarly showed stronger declines south than north (Matiu et al., 2021). Consequently, this trend might continue in the future given the findings in this study.

The downscaling introduced some bias at elevations below 1500 m, while above the procedure is largely unbiased (Figure S11b and c, left panel). But even at the lower elevations, the bias was lower than the model spread and future change estimates. Thus, the largest part of uncertainty of future projections was less because of the downscaling method, but more caused by the spread in GCM forcing together with RCM snow schemes. Since there is no single "best" climate model (Vautard et al., 2021) and no single best snow model (Etchevers et al., 2004; Rutter et al., 2009; Menard et al., 2021), we conclude it is safe to take model spread as representative of model uncertainty for future projections.

The employed statistical downscaling method extrapolates information beyond the elevation coverage of the RCMs. At 0.11°, and not considering the grid cells with snow accumulation, the highest grid cell from low-resolution RCMs was at approximately 3000 m, which contained single high-resolution pixels with elevations up to 4105 m. The downscaled estimates above 3000 m thus should be treated with caution, even though the observed stronger reductions at elevations above 3000 m from downscaling are similar to the results of the simulations from a high-resolution RCM, which explicitly resolved elevations up to 3500/4000 m (Lüthi et al., 2019).

A benefit of the proposed downscaling approach is that is based on the truly local features, which were derived from 20 years of observations. In contrast, the final imputation step of SNCp50 is based on a simple elevational dependence of snow cover, and could thus directly be estimated from a low-resolution RCM signal. At least, the initial derivation of SNCp50 and the first imputation step can be assumed to provide downscaled estimates based on local features, while the results of pixels that were subject to the final elevation imputation are more generalized. On the other hand, applying the downscaling grid cell by grid cell introduced artefacts at the low-resolution grid cell boundaries (see, e.g., Figure 5c). For the future, other downscaling techniques could be explored, such as analogue, perfect prog, or weather typing methods (Zorita and Storch, 1999; Gutiérrez et al., 2013), as well as, spatially explicit gridded downscaling approaches (Werner and Cannon, 2016).

One assumption in the downscaling is that the remotely sensed observations from MODIS are true, but these also have errors and noise. Generally, accuracies in determining binary snow information (snow or land) are largely above 90% for MODIS (Parajka and Blöschl, 2006; Gafurov and Bárdossy, 2009). However, considerable uncertainty and lower accuracies were found for forested areas and locations affected by terrain shading (Notarnicola et al., 2013b). Specific to this study is the use of a cloud filtered product, which provides gap-free spatiotemporal series. The used filtering techniques resulted in only slightly lower overall accuracies of 91.5% compared to 93% for the original images (Matiu et al., 2020a). However, the spatial and temporal filters that were applied to remove clouds might miss short snow episodes at low elevations and are difficult to validate at higher elevations, because of low ground station coverage. A pixel with erroneous information from

MODIS will translate to an erroneous downscaled pixel, so relying on single pixels without consulting the spatial surrounding is not advised.

In addition, the downscaling assumes no land cover change, which might be problematic, for example, where the tree line increases, and forests migrate to higher elevation. This comes on top to the already challenging estimation of snow cover fraction from remote sensing for forested areas. Under a warming climate, complex vegetation-snow interactions can occur,

such as opposing effects on the interception and subsequent melting of snow in forests (DeBeer et al., 2021).

## 4.3 Comparison of downscaling to a dedicated snow model

For the Ötztal Alps region in Austria (Figure 1), we compared results from bias adjustment and downscaling of RCM snow cover fraction (SNC) to running a dedicated snow model (AMUNDSEN), which has been forced by output from RCMs. For

the past period (2001-2020), the downscaling resulted in lower SNC than AMUNDSEN up to approximately 2000 m, similar SNC from 2100 to 2600m, and higher SNC for elevations above 2700 m (Figures S13 and S14). However, elevations above 2700 m are challenging to compare, since many pixels were removed from bias adjustment and downscaling at these elevations because of snow accumulation issues and glaciers, while AMUNDSEN resolved the whole domain and explicitly considered ice-snow transitions. Consequently, comparisons above 2700 m are not based on the same pixels.

The change estimates for the future period (2071-2100) under RCP8.5 agreed between bias adjustment, downscaling, and AMUNDSEN for elevations between 1800 and 2800 m, considering model ensemble uncertainty (Figure 6). But strong disagreement was observed above and below. For elevations below 1500 m, AMUNDSEN showed much stronger reductions in SNC than downscaling. The elevation gradient of projected changes under RCP8.5 differed substantially between AMUNDSEN and downscaling (Figure 6). While AMUNDSEN showed mostly constant absolute change across elevation,

with slightly stronger decreases between 1500 and 2000 m under RCP8.5, the bias adjusted or downscaled SNC from RCM showed a strong elevational gradient, such that absolute decreases in SNC became stronger with increasing elevation. In relative terms, AMUNDSEN had the highest change rates at lowest elevations and lowest change rates at highest elevations, while for downscaling the opposite was true.

This elevation gradient in the relative SNC changes from downscaling for the Ötztal Alps is counter-intuitive. It is also

different from the gradients for bias adjustment and dowscaling for the whole study area, which themselves are similar to the results from AMUNDSEN for the Ötztal Alps (Figure 6). One reason for this discrepancy might be that the Ötztal Alps region comprises only 15 RCM grid cells with a very limited elevation range (1800 to 2800 m), which has to be extrapolated to a much wider elevation range (900 to 3700 m) in the finer spatial resolution. In the case of AMUNDSEN, this extrapolation is performed on the surface meteorology, which seems to work better than the extrapolation performed in the

SNC downscaling approach.

A further cause of the strong differences in SNC changes especially at lower elevations might be due to the consideration of forest snow processes in the AMUNDSEN simulations, where a canopy submodule accounts for the interception of snow by

the trees – from where the snow can subsequently sublimate or melt without reaching the ground – as well as the modification of the meteorological variables for sub-canopy conditions; for details see (Strasser et al., 2011). As the

500 AMUNDSEN SNC results considered in this study only correspond to snow on the ground, this can cause differences with the RCM-based SNC changes, considering the large proportion of forested areas in the affected elevation bands (61 % forest coverage for elevations < 2000 m compared to only 2 % for elevations >= 2000 m).

Given this study's setup, it's not possible to disentangle how climate change signals and uncertainties flow through the modelling chain of both approaches with their different statistical post-processing and physical models. But we propose that

such an assessment would be beneficial for highlighting important aspects of the modelling uncertainty of future mountain snow cover. A related issue is that for the bias adjustment of SNC in this study RCMs caused more of the overall variability than their driving GCMs, while in Hanzer et al. (2018) it was the opposite. It seems that, for a small high-elevation area such as the Ötztal Alps, the RCMs cannot demonstrate their full potential and the large-scale forcing from GCMs takes precedence.

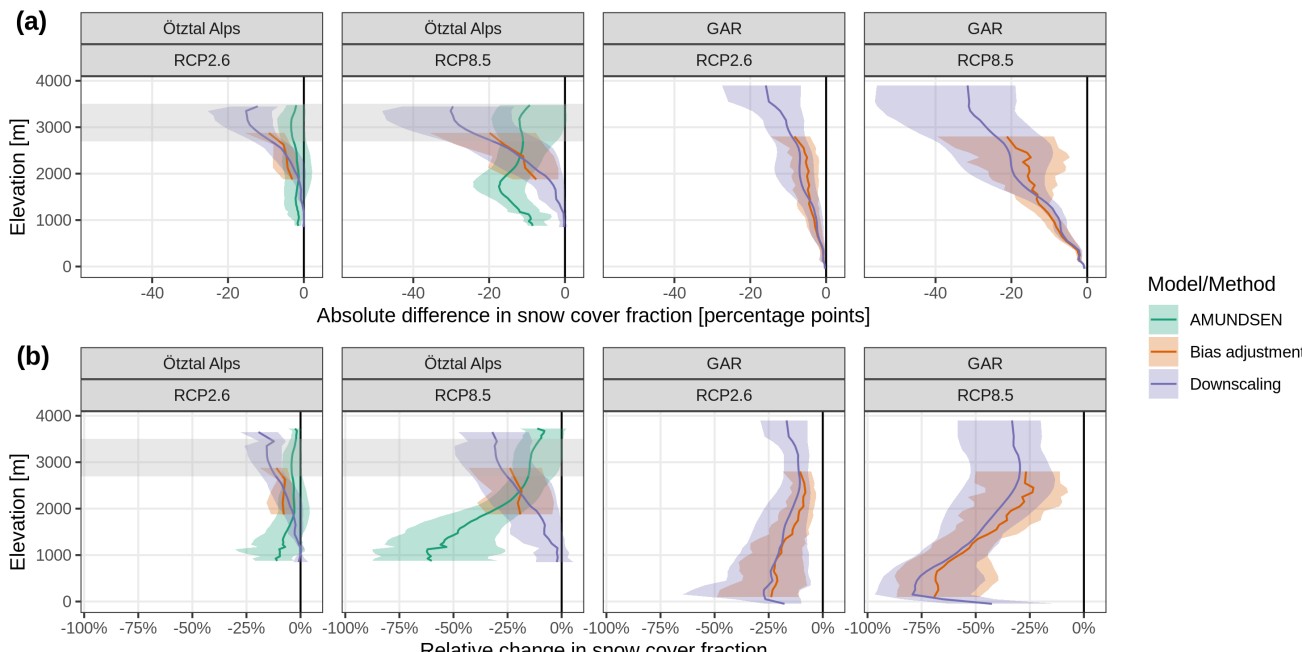

**Figure 6. Change in future snow cover fraction (SNC) for the Ötztal Alps region and the whole study area (GAR, Greater Alpine Region) by elevation band. Colored lines and transparent regions denote model means and model spread from running a snow and hydroclimatological model (AMUNDSEN), forced by downscaled meteorology from regional climate models (RCMs), from bias adjusted SNC from RCMs, and from downscaled SNC from RCMs. Shaded grey area in the Ötztal Alps panels (above 2700 m)**

**indicates elevations, where >20 % of the pixels entering the average per elevation band were removed from MODIS and Downscaling but remained included in AMUNDSEN: these consist of glacierized pixels or pixels subject to snow accumulation in RCMs, while AMUNDSEN resolved the whole domain. (a) shows absolute changes and (b) relative changes.**

To conclude the comparison of bias adjustment and downscaling to using a dedicated snow model, Table 1 offers an overview of their main features. Both approaches enable to assess climate model uncertainty by using model ensembles. Both suffer from the potential need to extrapolate the RCM signal (surface meteorology or snow cover) beyond its elevation coverage, especially in complex mountain terrain. Both approaches decouple surface meteorology and snow cover in the climate change signal. The main differences between the two approaches are in their spatial extent, spatial detail, the

representation of snow and ice processes, and the availability of observations.

**Table 1. Non-exhaustive comparison of benefits (denoted with a +) and drawbacks (denoted with a –) of the two methods considered in this study. Abbreviations: regional climate model (RCM), snow cover fraction (SNC).**

| Dedicated model forced by RCM meteorology | Bias adjustment (and downscaling) of RCM SNC |
|---|---|
| + Strong local (topographic) detail possible | – Limited by RCM resolution; artefacts at grid cell boundaries |
| + Detailed representation of snow and ice processes | – Limited by adequacy of RCM snow scheme |
| + All snow cover variables (water equivalents, depths, area covered) | – Only snow cover fraction |
| – Limited spatial extent | + Applicable at large spatial scales |
| – Requires surface meteorology (in situ data, downscaling of RCM output) | + Observations from remote sensing globally available |
| (–) Requires extensive snow modelling experience | (+) Mostly statistical and computational skills required |

Previous studies on the future of snow cover in the European Alps found differing trend magnitudes, but quantitative comparisons are hampered by different study extents and emission scenarios.

Marty et al. (2017) found decreases of snow cover duration until the end of the century from 100 to 14-18 d at 1000 m, 157 to 49 d at 1500 m, and 254 to 163 d at 2700 m (cf. Table S2 in Marty et al. (2017)), while here we found reductions that were

535 much lower: 30, 47, and 82 d at the respective elevations (Table S8). Their estimates were based on the Alpine3D snow model for subregions of Switzerland, forced by RCM meteorology from the ENSEMBLES project, under the A2 emission scenario, which has less GHG concentrations at the end of the century compared to the RCP8.5 in this study. The differences in change estimates might partly be caused by different reference periods for the past.

Lüthi et al. (2019) found a decrease of 60% in SWE and a two months shorter snow cover duration by analyzing one

regional climate model at 2 km spacing over the Alpine region under RCP8.5, while we observed a 49% reduction in SNC and an average reduction of 22 d in SCD. Trend differences might be explained by the fact that domain averages strongly depend on the investigated domain, and Lüthi et al. (2019) have a different extent of Alpine region compared to this study. In addition, their domain includes much higher elevations because the horizontal spacing is much finer, and the higher elevations showed stronger reductions in snow cover.

**4.4 Validation of the downscaling**

The downscaling approach was validated by applying it to the upscaled MODIS snow cover fraction, and then compare it to the original maps, which have been used for upscaling. This comparison involved the whole domain and all daily maps, resulting in approximately 71 billion pixels (7305 days times 9.8 million pixels per map). The overall accuracy of the

550 downscaling was 96.4%, the PPV 89.1%, and the NPV 97.4%. Consequently, snow was downscaled less correctly than land, but still accuracies are high. In addition, there was a seasonal and elevational dependence of the downscaling errors. Lowest accuracies were found for the elevation at which the transition from land to snow occurs, and this elevation varied by season (Figure S15). For example, in December, the lowest accuracy was 83 % at1400 m, but in May, the lowest accuracy was 87 % at 2000 m. In absolute terms, the number of correctly downscaled pixels outweighs the errors by large (Figure S16).

To evaluate the errors in downscaled climatologies of SCD, we compared the downscaled QDM bias adjusted past RCM to the observed high-resolution climatology from MODIS. By definition of QDM, the empirical distribution of past snow cover fraction at the 0.11° resolution is identical between RCMs and MODIS. Thus, the difference between downscaled average annual snow cover duration (SCD) and observed high-resolution MODIS SCD is an indicator of the downscaling error. The mean downscaling bias was -3.0 d and the MAE 5.2 d. In addition, there was an elevation dependence of the bias (Figure

S17). A negative bias was found for elevations below 1000 m, almost no average bias between 1000 and 3000 m, and positive bias above 3000 m. Glacierized surfaces exhibited strong positive bias, except if SNCp50 was imputed by the second elevational step.

The downscaling procedure assumes seasonal stationarity. Across the snow season, the processes governing snow accumulation and ablation differ substantially, so seasonal stationarity is questionable. However, the downscaling procedure

employed in this study is based on terrain morphology, which stays constant across the season. For the spatial scales used here with 250 m spacing, this resolves to mostly elevation and only partly aspect and slope. For higher spatial spacings, such as tens of metres, preferential deposition of snow, terrain shading, and wind start to play strong roles, and stationarity becomes increasingly less plausible. We evaluated the stationarity assumption for our study by calculating two different SNCp50 values, one for the start of season (September to February) and one for the end of season (March to August). The

average bias between the seasonal and annual SNCp50 values was -0.018 for the start and 0.007 for the end of season, with an MAE of 0.057 and 0.041, respectively. Most of the bias was confined to lowest elevations (below 1000 m), which do not have a proper start and end of season, but multiple intermittent episodes. Given these low differences between seasonal SNCp50 values, the seasonal stationarity assumptions seems justified.

## 5 Conclusion

Bias adjustment of snow cover fraction from RCMs using aggregated MODIS remote sensing observations offers a promising approach to evaluate future changes of snow cover fraction in mountain areas under a climate model ensemble view. While limited by the resolution of RCMs, it offers consistent large-scale patterns of snow cover fraction for the past and future, and it is potentially applicable on a global scale. Consequently, it might be a viable alternative in remote or less monitored areas. While snow cover fraction is probably not a priority for climate modelling groups to be made available, the proposed bias adjustment could benefit from a larger ensemble of climate models being available. Regarding bias adjustment methods, trend-preserving approaches, such as delta change or quantile delta mapping, were found superior to quantile mapping for snow cover fraction.

The downscaling of RCM SNC with high-resolution MODIS observations falls under an "experimental" label. It suffers from many inadequacies, such as snow accumulation in RCMs, noise in observations, and is likely inappropriate for glacierized areas. However, it can provide auxiliary and high spatial resolution information while accounting for climate model uncertainty. The discrepancies to results from a dedicated snow model for a smaller area (the Ötztal Alps) require further research before a final recommendation can be given. Even though differences are likely caused by the small spatial extent and catchment specifics (forest distribution, glaciers), which are features that fall outside of the RCM's scope and capabilities.

For the study region, which is approximately the Greater Alpine Region, results showed an overall reduction in snow cover fraction for 2071-2100 compared to 2001-2020 of 14 % for RCP2.6 and 48 % under RCP8.5. However, strong elevational and seasonal dependencies of changes were found (Tables S3 to S8, Figures S8, S9, S11). Absolute reductions became higher with increasing elevation, while relative reductions became lower (Table 2). Downscaling resulted in slightly more negative estimates of change than solely from bias adjustment. In addition, spatial patterns of change emerged, with stronger relative decreases in the south and west compared to north and east (Figure 5), which are consistent to past trends of station observations of snow depth (Matiu et al., 2021). Results for the low emission scenario RCP2.6 are based on a smaller ensemble than for the high emission scenario RCP8.5 (4 vs. 23 models), and thus model uncertainty might be underestimated for RCP2.6.

Potential usages of the downscaled information include hydrological studies or glacier modelling studies that require snow line information. They might help in determining winter sport reliability, even though future assessments that do not account for technical snow are most likely not very useful (Spandre et al., 2019; Morin et al., 2021). Finally, downscaling approaches should be kept in mind considering the new generation of soon to be available high-resolution RCMs (at or below 2 km), for example, from CORDEX flagship pilot studies, together with long-term remote sensing observations at tens of meters scale, such as harmonized Landsat Sentinel series.

**Table 2. Summary of changes in snow cover fraction (BA, bias adjustment) and snow cover duration (DS, downscaling) by emission scenario at three representative elevations. Columns show model mean with model spread in parentheses for absolute (abs.) changes in percentage points (pp) for BA, days for DS, as well as, relative (rel.) changes for BA and DS. RCP stands for Representative Concentration Pathway. Results are based on quantile delta mapping as BA method.**

| Scenario | Elevation [m] | BA abs. [pp] | BA rel. | DS abs. [d] | DS rel. |
|---|---|---|---|---|---|
| RCP2.6 | 500 | -1.5 (-2.6, -0.5) | -21.2% (-37.0, -7.9) | -5 (-9, -1) | -23.6% (-44.3, -6.1) |
| | 1,500 | -4.4 (-7.7, -2.0) | -14.0% (-24.8, -6.5) | -18 (-32, -7) | -18.1% (-32.6, -7.6) |
| | 2,500 | -5.8 (-9.9, -2.7) | -8.4% (-14.5, -4.0) | -26 (-47, -14) | -10.7% (-19.5, -6.0) |
| RCP8.5 | 500 | -4.7 (-6.0, -2.7) | -68.6% (-87.1, -39.6) | -16 (-19, -10) | -76.3% (-93.0, -45.9) |
| | 1,500 | -13.2 (-19.0, -9.0) | -42.3% (-60.9, -28.9) | -47 (-65, -35) | -48.1% (-66.9, -36.3) |
| | 2,500 | -17.2 (-32.8, -6.4) | -25.2% (-48.0, -9.4) | -76 (-134, -33) | -31.4% (-55.3, -13.7) |

**Code and data availability**

All code to perform the analysis is available in a public repository (Matiu, 2021). The repository also holds final processed data of the snow cover duration climatologies from MODIS, as well as single GCM-RCM maps from bias correction and

downscaling. The input data is not shared, because of its large size. The RCM data is available for non-commercial use after registration (https://cordex.org/data-access/). For access to the MODIS observations, see Matiu et al. (2020a). The AMUNDSEN data is available from FH upon request.

**Author contribution**

MM defined the study concept, performed the formal analysis, and wrote the original draft. MM and FH were involved in

data curation. Paper review and editing were performed by MM and FH.

**Competing interests**

The authors declare that they have no conflict of interest.

**Acknowledgements**

This project has received funding from the European Union's Horizon 2020 research and innovation programme under the Marie Sklodowska-Curie grant agreement no. 795310.

We acknowledge the World Climate Research Programme's Working Group on Regional Climate, and the Working Group on Coupled Modelling, former coordinating body of CORDEX and responsible panel for CMIP5. We also thank the climate modelling groups for producing and making available their model output. We also acknowledge the Earth System Grid Federation infrastructure an international effort led by the U.S. Department of Energy's Program for Climate Model Diagnosis and Intercomparison, the European Network for Earth System Modelling and other partners in the Global Organisation for Earth System Science Portals (GO-ESSP).

We thank Valentina Premier for assistance and discussion regarding the downscaling procedure, and Marc Zebisch for general discussions.

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
