# Peer review of "Bias adjustment and downscaling of snow cover fraction projections from regional climate models using remote sensing for the European Alps"

_Hydrology and Earth System Sciences, 2021_

## Referee Comment (RC2)

**Summary and general comments**

In this study, a statistical bias adjustment and downscaling method for snow cover fraction data is explored. Simulated snow cover fractions are obtained from the EURO-CORDEX regional climate model ensemble at a horizontal resolution of 0.11° – both for the present day climate and for the end of the 21st century (for RCP 2.6 and 8.5). The bias adjustment is performed with MODIS remote sensing data, which is spatially aggregated from a resolution of 250 m to a similar scale as the EURO-CORDEX data. For the bias adjustment, four different methods are initially applied – delta change, quantile mapping, quantile delta mapping (QDM) and multivariate QDM. Bias-adjusted snow cover fractions are then subsequently downscaled to the initial high resolution of 250 m. The entire procedure is, for a subdomain, compared to a rather conventional approach in which snow cover fraction is derived from a high-resolution snow model, which is forced by downscaled RCM output.

This study explores an interesting and novel approach – namely the application of bias adjustment for snow cover fraction. The authors evaluate the suitability of different bias adjustment methods for this challenging approach. The manuscript is predominantly well written – some sections, like the explanation of the downscaling method – are however difficult to follow and should be improved. At some places, a more extensive discussion of the results would be useful – for instance regarding projected relative changes in snow cover fractions – which are surprisingly largest at high elevations.

**Major comments**

**Remapping of high-resolution MODIS and CORDEX data**
I'm a bit confused about the performed remapping. You first aggregate MODIS data to a low resolution grid and then remap CORDEX data to this grid with a nearest neighbour method, right? This seems to be unnecessarily complicated. Wouldn't it be easier to remap MODIS data (for instance with a conservative method) directly to the default rotated latitude/longitude CORDEX grid?

**Downscaling method**
The downscaling method is very interesting but difficult to follow in some places:
- A general question (just out of curiosity): downscaling can also be performed directly within the bias adjustment method. However, it seems to be less appropriate if observational data is available on a much higher spatial resolution. Did you nonetheless consider this option?
- Line 229: Is the requirement for the function not rather "strictly increasing" to get a unique solution for SNCp50? Furthermore, I don't understand the subsequent part with the "longest non-strictly increasing subsequence"
- I found it particularly difficult to follow the "filling of missing SNCp50 values" section (lines 235 – 248). The comprehensibility might be improved with an additional sketch or figure. Moreover, I have some specific questions:
  - Could you explain why the Wasserstein distance is a suitable metric for this problem?
  - What do you mean by "no one-to-one correspondence"?
  - By nearest low-resolution pixels (line 240), you refer to the Wasserstein distance, not the horizontal distance, right?
  - Line 247: how do you determine these 100 pixels exactly from the theoretically 50 * 100 available pixels?
  - How did you proceed with the very small fraction (<0.001%) of pixels with still missing SNCp50 values?
- Line 249: Could the downscaling approach not simply be evaluated by reconstructing high-resolution MODIS snow cover from the spatially aggregated pixels?
- It seems that you assume a seasonally stationary downscaling relation, i.e. there is no temporal dependency. Is this assumption valid? Or could the relation look slightly different for e.g. late autumn and spring?

**Projected changes in snow cover fraction for different elevations**
I'm surprised that the elevation gradients in Figure S12 are so distinctively different. Could this also be caused by the different RCM ensembles used for AMUNDSEN and the statistical approach (i.e. do different RCMs for instance show different elevation gradients in warming)? It is generally a bit counter-intuitive that the largest relative reduction in snow cover fraction is projected for the highest elevations (see Figure S10b and S12).

**Minor comments**

**Content-related (text)**
**Line 15:** With which bias adjustment method were these number computed?
**L17:** "The comparison of the statistical" → This evaluation confirms the robustness of the downscaling method
→ I would therefore mention it before presenting results for the future climate.
**L19:** The term "bias correction" is a bit outdated. Better use the term "bias adjustment" (see e.g.
https://cordex.org/data-access/bias-adjusted-rcm-data/ and https://hypeweb.smhi.se/what-is-bias-adjustment/)
**L19:** "plausible" might be a bit vague. Maybe better "more suitable" or "applicable".
**L37:** I find it difficult to follow this sentence. Maybe it's better to write "because climate change violates the assumption of stationarity" (if you mean that).
**L53:** "…and even snow depth" → this part seems to be a bit contradictory to the previous sentence. Snow depth is reproduced relatively well but SWE not?
**L54:** "gridded estimates of SWE" → You refer to data derived from in-situ measurements (and not models), right? If so, I would explicitly state this. Furthermore, I guess you want to allude to the spatial representativeness of different data sets (in-situ vs. remote sensing) here. If so, you could discuss this in an additional sentence here…
**L61:** "The reference observations can be points or grids, are often limited in extent compared to RCMs, and feature, in case of grids, typically higher resolutions."
**L82:** "The motivation is that…" → this sentence is rather long and difficult to read – could you rephrase it? I furthermore do not understand the end "…while future change estimates should be consistent." Consistent with what?
**L84:** "While bias correction…" → I don't understand this sentence entirely. What is the link between absolute values and bias adjustment?
**L118:** Did you apply a nearly or entirely cloud-free data set in the end?
**L119:** What does "nominal" and "effective" resolution mean in this context?
**L127:** "all variables" → which variables do you mean?
**L139:** I was wondering – would it also be possible to account for the snow accumulation issue within the bias adjustment method?
**L160:** What do you mean by "single GCM-RCM biases"?
**L171:** I would be careful that the reader is not confusing calendar with hydrological years. I guess you conduct all your evaluations for hydrological years – right? Then I would state the definition of a hydrological year somewhere and mention, that all years refer to hydrological years – unless otherwise stated.
**L175:** I would remove ", usually 1970 – 2000"
**L189:** "also to the maximum of 1." → this is oddly phrased. I guess values above 0.999 are set to 1.0, right?
**L188:** The trace condition was applied to the about of the Q(D)M algorithm - right?
**L195:** "breaks the temporal consistency in the bias corrected SNC time series" → what do you mean by that?
**L260:** What do you mean by "preferred"? Are trends only preserved for ratio variables?
**L388:** What do you mean by "analogue based"?

**Typos, phrasing and stylistic comments**
**General:** Both the terms grid cell and pixel are used. To avoid ambiguity, it would be good to use them consistently: You could for instance use grid cell exclusively for climate models and pixel for remote sensing data.

**L22:** remove "only"
**L28:** "consequences in water supplies" sound odd. Maybe rather "on"?
**L31:** maybe better "snow cover causes a significant atmospheric feedback due to its high albedo"
**L50:** "such as for" sounds a bit odd. Maybe better: "for instance in"
**L51:** "but also from biases in RCM snow cover." I would rephrase this to something like: "but also from biases caused by the relatively simple snow schemes of RCMs."
**L67:** I would rearrange this sentence to: "Since QM has been show to modify trends in a few cases, quantile delta mapping (QDM) was developed, which represents a trend-preserving QM approach."
**L77:** "We restrict the study to snow cover fraction, which is, in contrast to snow depth and SWE, globally available on a high spatial resolution. The presented method has therefore a global potential for application."
**L82:** "meteorology" → "output"

**L88:** I would replace "representation" by "pattern". "Representation" can be ambiguous in this context (one could understand that snow cover representation in RCMs is improved).
**L91:** "is as" → "is structure as"
**L95:** The study region (Fig. 1) encompasses the European Alps and spans from ~43 – 48.5° N and ~5 – 17° E, which roughly corresponds to the Greater Alpine Region (Auer et al., 2007).
**L133:** "although snow depth match very good" → "although snow depths are well reproduced"
**L172:** "merged to" → "merged with"
**L270:** "under- and overestimating" → "simulated"
**L305:** change "lower ensemble" to "smaller ensemble"
**L325:** change "a more of a fully snow free and fully snow-covered grid cell over time" to "more often fully snow free or snow-covered conditions over time"
**L363:** I would not write "became" if you compare two hypothetical projections
**L383:** Change to: "In contrast, the final imputation step of SNCp50 is based on a simple elevational dependence of snow cover, and could thus directly be estimated from a low-resolution RCM signal."
**L396/397:** "in low/higher elevations" → "at low/higher elevations"
**L407:** "meteorology" → "output" or "meteorological output"
**L470:** What do you mean by "two months short snow cover duration"? A shortening by two months?
**L474:** "lower horizonal spacing" → "higher horizontal spacing"

**Figures and Tables**
**Figure 3:** Why is there no line in the upmost row in column "DC"? Furthermore, are results shown in this figure computed with bias adjustment and downscaling (or only the former)?
**Figure 6:** Interestingly, the line for "bias correction" is closer to "AMUNDSEN" than "downscaling" in the rightmost panel. Can you explain this?

**Table 1:** I'm only partially convinced about the point "+ Consistent climate change signal of surface meteorology with snow cover". This is true for the raw RCM output, but no longer for the bias corrected product.

**Figure S1:** Replace "snow plausible" in caption with e.g. "snow accumulation"
**Figure S2:** "surrounding" is a bit misleading in the caption – maybe better "aggregated"
**Figure S11:** caption line 2: "and downscaling."
**Figure S12:** I guess the part in the squared brackets belongs to the figure caption?

**Table S5:** I'm not sure if the reference "See also Figure S6" is correct/intended.

---

## Author Comment (AC1)

This paper used a bias correction and downscaling method to estimate snow cover fraction from regional climate models. The method and results are difficult to follow. Comments and questions are below.

Thank you for the review and constructive feedback.

We can elaborate more and better on the methods and results in a revised version. As the other referee has also pointed out that the methods are difficult to follow, we shall enhance readability and provide additional figures and schemas to make the procedure easier to understand.

1. Figure 1, the outline of the small study area is not clear. Please also give the description of the small study area. In Figure c and d, which data source does the snow cover duration come from?

We can improve Figure 1 in a revised version. The snow cover duration is from MODIS, based on the same data used in the downscaling procedure. We can clarify this in the caption.

2. L95-100 belong to "Study Area", rather than "Data".

We can add a separate section titled "Study Area".

3. Section 2.1 "Observed snow cover fraction from remote sensing", in addition to pointing to referenced paper (Matiu et al., 20201), state how you obtained the daily cloud-free snow cover maps?

The cloud-free maps were obtained by a series of temporal and spatial filters.

1. We applied a mean filter over a square of size 299 pixels centered on each pixel, where a snow pixel was reclassified as clouds, if the neighborhood contained more cloud than snow pixels, and a cloud pixel was reclassified as snow, if the neighbourhood contained more snow than cloud pixels. This was performed to improve accuracy at cloud borders or within clouds, where misclassifications sometimes occurred. This step was only performed for April to October, where this issue was mostly prevalent.
2. Then, we applied a conservative temporal filter, which is based on the persistence of snow. A cloud pixel was reclassified as snow, if it was snow one or two days before, as well as snow one or two days after. The same for land pixels.
3. Afterwards, a spatial filter was applied based on snow and land lines, which exploits the elevational dependence of snow cover. For any given day, all cloud pixels above the average elevation of all snow pixels were reclassified snow, and all cloud pixels below the average elevation of all land pixels were reclassifed as land. This step was

not applied if the image had more than 50% clouds (-> snow and land lines not representative), the ratio of snow to land pixels was less than 0.05 (-> too little snow for snow line to be representative), and not in the summer months June to September (-> too little snow)

4. Finally, a greedy temporal filter was applied that replaced any cloud pixel with the next available observation (land or snow) from either forward or backward in time. Most values were filled after a few days, but we set a maximum of ten days as the threshold to look for cloudfree observations.

We can provide a summary of these steps in a revised version.

4. Change "2.1 Snow cover fraction from regional climate models" to "2.2 Snow cover fraction from regional climate models".

Thanks for spotting this error.

5. L155-157, please provide more detailed information for the conversion from snow water equivalent to snow cover fraction. Snow cover fraction is a ratio, how did you use the threshold of 5 mm to get the snow cover fraction.

We used the 5mm threshold to create a binary value for each pixel/grid cell and each day. This binary value was then either snow or snow-free. Snow cover fraction was then calculated either along space, time, or spacetime by calculating the ratio of pixels with snow divided by all pixels.

6. L162,why use 19 GCM-RCM for RCP8.5 and 3 GCM-RCM for RCP2.6?

This imbalance between RCP8.5 and RCP2.6 scenarios results from the fact that more RCP8.5 simulations are available than for any other scenarios. We also tried to take only the common GCM-RCM pairs, but found that results for the mean were similar, and more GCM-RCMs give a better estimate of the modelling spread. So, we decided to keep the maximum available number. We can elaborate on this further in a revised manuscript, and also make the hint to the little number of scenarios for RCP2.6 more prominent.

7. L169-170, the full name of DC, QM, QDM have provided in the Introduction section, here just use DC, QM, QDM.

Fine.

8. How did you validate the estimated snow cover fraction?

We intended to use the comparison between the physical-based AMUNDSEN model and the statistical-based downscaling as a kind of two-way validation for both approaches. Since these are projected changes, we have no real validation data set, but can only perform comparison betweens approaches and models. For example, the comparison between raw, bias corrected, and downscaled estimates from RCMs as well as AMUNDSEN can give some hints on the differences and agreements. But without a true validation data set, there is little else possible.

On the other hand, we validated the downscaling procedure internally, by applying it to the upscaled snow cover fraction from MODIS (L249ff). We can provide more information on this in a revised manuscript.

Otherwise, the general capability of RCM to reproduce snow cover fraction can be validated for the past. Such an evaluation/validation has been performed previously in Matiu et al. 2020b, mentioned in the introduction, (https://doi.org/10.3390/atmos11010046) using most of the GCM-RCMs in this study and the similar MODIS observation. We can provide more information on this previous validation in a revised version.

---

## Author Comment (AC2)

**Summary and general comments**

In this study, a statistical bias adjustment and downscaling method for snow cover fraction data is explored. Simulated snow cover fractions are obtained from the EURO-CORDEX regional climate model ensemble at a horizontal resolution of 0.11° – both for the present day climate and for the end of the 21st century (for RCP 2.6 and 8.5). The bias adjustment is performed with MODIS remote sensing data, which is spatially aggregated from a resolution of 250 m to a similar scale as the EURO-CORDEX data. For the bias adjustment, four different methods are initially applied – delta change, quantile mapping, quantile delta mapping (QDM) and multivariate QDM. Bias-adjusted snow cover fractions are then subsequently downscaled to the initial high resolution of 250 m. The entire procedure is, for a subdomain, compared to a rather conventional approach in which snow cover fraction is derived from a high-resolution snow model, which is forced by downscaled RCM output.

This study explores an interesting and novel approach – namely the application of bias adjustment for snow cover fraction. The authors evaluate the suitability of different bias adjustment methods for this challenging approach. The manuscript is predominantly well written – some sections, like the explanation of the downscaling method – are however difficult to follow and should be improved. At some places, a more extensive discussion of the results would be useful – for instance regarding projected relative changes in snow cover fractions – which are surprisingly largest at high elevations.

Thank you for the positive appreciation of our work and your extensive review, which can greatly improve a revised version of the manuscript.

**Major comments**

**Remapping of high-resolution MODIS and CORDEX data**
I'm a bit confused about the performed remapping. You first aggregate MODIS data to a low resolution grid and then remap CORDEX data to this grid with a nearest neighbour method, right? This seems to be unnecessarily complicated. Wouldn't it be easier to remap MODIS data (for instance with a conservative method) directly to the default rotated latitude/longitude CORDEX grid?

Yes, your proposition seems at first easier. However, it has been complicated by computational constraints. The support for the rotated CORDEX grid is very low in the software we used, namely R statistical software. At the start of the project (2-3 years ago), it was complicated if not impossible to deal with the rotated grid in R appropriately. Nowadays, support is a little better, but still not good. So, we decided to take the more standard LAEA grid of MODIS as backbone and reproject RCMs.

**Downscaling method**
The downscaling method is very interesting but difficult to follow in some places:
  • A general question (just out of curiosity): downscaling can also be performed directly within the bias adjustment method. However, it seems to be less appropriate if observational data is available on a much higher spatial resolution. Did you nonetheless consider this option?

Yes, our initial idea was also to directly apply QM or QDM with observational data. In our case, however, we had to apply it to different types of variables: At 0.11° RCM scale we have a continuous but bounded snow cover fraction (0-100%) and at 250m observation we have a binary snow/land variable. So, we could not apply QM and QDM directly. On a side note: While the MODIS product we used is a binary one that has been developed at the institute, there exist also other options. Eg. the MODIS V6 product from NASA offers the NDSI (normalized difference snow index), from which snow cover fraction could be calculated. So with other observational data, the bias adjustment could be combined with downscaling directly. This would be an interesting alternative, but beyond the scope of this study...

  • Line 229: Is the requirement for the function not rather "strictly increasing" to get a unique solution for SNCp50? Furthermore, I don't understand the subsequent part with the "longest non-strictly increasing subsequence"

Actually, non-strictly increasing suffices. SNCp50 occurs when the y=0.5 line is crossed (e.g. Figure 2). For

this, it is not important if we have a flat (non-increasing) portion. What destroy the uniqueness is when the y=0.5 is crossed twice or more times, which happens if the curve is decreasing at some point.

Regarding the second point, if the curve is decreasing at some point or even at multiple points, then there are multiple choices to select a monotonic subsequence. So we selected the longest possible of these, assuming it is the best estimate. We can provide an additional explanatory figure in a revised manuscript, if required.

- I found it particularly difficult to follow the "filling of missing SNCp50 values" section (lines 235 – 248). The comprehensibility might be improved with an additional sketch or figure. Moreover, I have some specific questions:
  - Could you explain why the Wasserstein distance is a suitable metric for this problem?
  - What do you mean by "no one-to-one correspondence"?
  - By nearest low-resolution pixels (line 240), you refer to the Wasserstein distance, not the horizontal distance, right?
  - Line 247: how do you determine these 100 pixels exactly from the theoretically 50 * 100 available pixels?
  - How did you proceed with the very small fraction (<0.001%) of pixels with still missing SNCp50 values?

That is a good idea. We can provide a graph/figure in a revised manuscript that gives an overview of the procedure, and we can also improve the writing in general. We can clarify all of the below points in a revised manuscript.

Regarding your specific points:
- The Wasserstein distance works for two sets of values that do not have to have a one-to-one correspondence between observations, such as, for example, probability distributions. The elevations of high-res grid cells can be considered such a "probability distribution". If we used other metrics, such as Euclidean distance, then we need to have one-to-one correspondence between the high-res elevations of two low-res grid cells. Such a correspondence could be e.g. by always comparing the topleft high-res grid cell of low-res grid cell A with the topleft high-res gridcell of low-res grid cell B, … , and the bottomright to the bottomright. But this would create not useful distances for the task, which is to compare the sub-grid elevation distributions.
- One-to-one correspondence: see previous comment.
- Exactly, at line 240 we refer to the Wasserstein distance.
- At line 247, we order the pixels by elevation difference.
- The rest of the <0.001% pixels were removed from the analysis.

- Line 249: Could the downscaling approach not simply be evaluated by reconstructing high-resolution MODIS snow cover from the spatially aggregated pixels?

That is true. We could also apply it directly to the upscaled observed MODIS data. We can explore this analysis in a revised version.

- It seems that you assume a seasonally stationary downscaling relation, i.e. there is no temporal dependency. Is this assumption valid? Or could the relation look slightly different for e.g. late autumn and spring?

Yes, we assume a seasonally stationary downscaling relation. The main reason is that the main components of change in snow cover fraction are snow deposition driven by snowfall and snow disappearing from snow melt. The strongest element in both snowfall and melt is elevation - at least at the scales used here in the 250m range. And this is what the downscaling procedure extracts.

Once we go up to higher resolutions, such as 10-20m, where preferential deposition of snow, terrain shading, and wind start to play strong roles - then the seasonal stationarity is questionable, since it will strongly depend on whether it's the start of the season or end of the season.

One idea to test the seasonal stationarity assumption is to derive two different SNCp50 values, one for the first half-year (e.g. Aug-Jan) and one for the second (Feb-Jul), and see whether they differ systematically.

**Projected changes in snow cover fraction for different elevations**

I'm surprised that the elevation gradients in Figure S12 are so distinctively different. Could this also be caused by the different RCM ensembles used for AMUNDSEN and the statistical approach (i.e. do different RCMs for instance show different elevation gradients in warming)? It is generally a bit counter-intuitive that the largest relative reduction in snow cover fraction is projected for the highest elevations (see Figure S10b and S12).

Yes, we too were surprised about the large differences in the elevation gradients in Figure S12. While beyond the scope of the study, it would be very interesting to understand where these differences come from. However, this would require a different experiment setup (same RCM-GCMs, with and without bias adjustment, …). Regarding your questions: Yes, different RCMs show different elevation gradients, see e.g. Fig S5 or S8. But while the model ensemble affects the spread, for the mean signal, the difference between the ensembles used for AMUNDSEN and the statistical approach should be little.
In Figures S10b and S12, we show the difference in percentage points, which is the absolute change in snow cover fraction, whose units are already percent. These are not true relative changes. In addition, the changes are for the annual snow cover fraction. While in winter largest changes are found for low-middle elevations, in spring, and likely summer too, the largest changes are for higher elevations - see e.g. Fig S5. In a revised manuscript, we could show a seasonal version of Fig S12, as well as a true relative change figure. These could then maybe explain better why the absolute changes are highest for higher elevation. We can also clarify better the difference between absolute changes in percentage points and relative changes in percent.

**Minor comments**

**Content-related (text)**
**Line 15:** With which bias adjustment method were these number computed?

QDM. We can report it in the abstract, too.

**L17:** "The comparison of the statistical"    This evaluation confirms the robustness of the downscaling method I would therefore mention it before presenting results for the future climate.

Makes sense.

**L19:** The term "bias correction" is a bit outdated. Better use the term "bias adjustment" (see e.g.
https://cordex.org/data-access/bias-adjusted-rcm-data/ and
https://hypeweb.smhi.se/what-is-bias-adjustment/)

Thank you for pointing this out. We shall change the terminology throughout the manuscript.

**L19:** "plausible" might be a bit vague. Maybe better "more suitable" or "applicable".

"More suitable" sounds better.

**L37:** I find it difficult to follow this sentence. Maybe it's better to write "because climate change violates the assumption of stationarity" (if you mean that).

That's what we meant. We can rewrite it.

**L53:** "…and even snow depth"    this part seems to be a bit contradictory to the previous sentence. Snow depth is reproduced relatively well but SWE not?

Yes and no. We could mention factors that might lead to these contradicting results, such as that in-situ snow depth observations are of generally higher quality than gridded SWE estimates, or that it could depend on the region.

**L54:** "gridded estimates of SWE"    You refer to data derived from in-situ measurements (and not models), right? If so, I would explicitly state this. Furthermore, I guess you want to allude to the spatial representativeness of different data sets (in-situ vs. remote sensing) here. If so, you could discuss this in an

additional sentence here…

The SWE grids we mention here are based on in-situ and remote sensing data, with some (empirical or physical) models behind. But we could also mention in more sentences the spatial issue.

**L61:** "The reference observations can be points or grids, are often limited in extent compared to RCMs, and feature, in case of grids, typically higher resolutions."

Thanks for the proposition. It sounds better.

**L82:** "The motivation is that…" this sentence is rather long and difficult to read – could you rephrase it? I furthermore do not understand the end "…while future change estimates should be consistent." Consistent with what?

Consistent across time. So, if we assume systematic biases, these should be consistent across time. We can rephrase and shorten the sentences.

**L84:** "While bias correction…" I don't understand this sentence entirely. What is the link between absolute values and bias adjustment?

Here we meant bias adjustment with downscaling. We can rephrase it.

**L118:** Did you apply a nearly or entirely cloud-free data set in the end?

It was a nearly cloud-free data set.

**L119:** What does "nominal" and "effective" resolution mean in this context?

Nominal is 250m, effective in the LAEA projection is 233m.

**L127:** "all variables" which variables do you mean?

While the coverage for temperature and precipitation is high, snow variables such as snow cover fraction are not available for all models.

**L139:** I was wondering – would it also be possible to account for the snow accumulation issue within the bias adjustment method?

We think not. If we assume a fully snow covered grid cell across the whole simulation period, this would likely render the change estimates for this grid cell inadequate. For SWE, the change would be too positive, for snow cover fraction, no change would occur. The bias adjustment might pull the hindcast closer to observations, but it cannot "correct" for this accumulation for the future.

**L160:** What do you mean by "single GCM-RCM biases"?

We meant that QM removes the biases for each GCM-RCM for the past period, and thus also the comparison for future estimates will be affected.

**L171:** I would be careful that the reader is not confusing calendar with hydrological years. I guess you conduct all your evaluations for hydrological years – right? Then I would state the definition of a hydrological year somewhere and mention, that all years refer to hydrological years – unless otherwise stated.

Actually, we performed calculations for the RCM data for calendar years. The bias correction has been performed month by month anyway and with distributions. Only for calculating snow cover duration from the incomplete MODIS data (Feb 2000 to Aug 2020), we used hydrological years. We can clarify this.

**L175:** I would remove ", usually 1970 – 2000"

Ok.

**L189:** "also to the maximum of 1." this is oddly phrased. I guess values above 0.999 are set to 1.0, right?

Exactly.

**L188:** The trace condition was applied to the about of the Q(D)M algorithm - right?

Yes.

**L195:** "breaks the temporal consistency in the bias corrected SNC time series" what do you mean by that?

The random component is added in a distributional manner for each month and 30 years of data. So for climatological averages for that month, that should be fine. But at the daily scale, this random component can cause, e.g., that even during the melting period snow cover fraction can increase. And also the year-to-year variability might not be representative.

**L260:** What do you mean by "preferred"? Are trends only preserved for ratio variables?

No, for QDM trends are preserved also for the interval variables. What we meant here was that, the trend-preserving attribute is preferred for ratio variables in general.

**L388:** What do you mean by "analogue based"?

We meant climate analogues, which is a downscaling technique that looks for similar (analogue) observations in the past that might be in different locations. We can explain this further in a revised manuscript.

**Typos, phrasing and stylistic comments**
**General:** Both the terms grid cell and pixel are used. To avoid ambiguity, it would be good to use them consistently: You could for instance use grid cell exclusively for climate models and pixel for remote sensing data.

Good idea. We shall employ consistent naming in a revised manuscript. We can also include all suggestion below.

**L22:** remove "only"
**L28:** "consequences in water supplies" sound odd. Maybe rather "on"?
**L31:** maybe better "snow cover causes a significant atmospheric feedback due to its high albedo" **L50:** "such as for" sounds a bit odd. Maybe better: "for instance in"
**L51:** "but also from biases in RCM snow cover." I would rephrase this to something like: "but also from biases caused by the relatively simple snow schemes of RCMs."
**L67:** I would rearrange this sentence to: "Since QM has been show to modify trends in a few cases, quantile delta mapping (QDM) was developed, which represents a trend-preserving QM approach." **L77:** "We restrict the study to snow cover fraction, which is, in contrast to snow depth and SWE, globally available on a high spatial resolution. The presented method has therefore a global potential for application." **L82:** "meteorology" "output"

**L88:** I would replace "representation" by "pattern". "Representation" can be ambiguous in this context (one could understand that snow cover representation in RCMs is improved).
**L91:** "is as" "is structure as"
**L95:** The study region (Fig. 1) encompasses the European Alps and spans from ~43 – 48.5° N and ~5 – 17° E, which roughly corresponds to the Greater Alpine Region (Auer et al., 2007).
**L133:** "although snow depth match very good" "although snow depths are well reproduced"
**L172:** "merged to" "merged with"
**L270:** "under- and overestimating" "simulated"
**L305:** change "lower ensemble" to "smaller ensemble"
**L325:** change "a more of a fully snow free and fully snow-covered grid cell over time" to "more often

fully  snow free or snow-covered conditions over time"

**L363:** I would not write "became" if you compare two hypothetical projections

**L383:** Change to: "In contrast, the final imputation step of SNCp50 is based on a simple elevational dependence  of snow cover, and could thus directly be estimated from a low-resolution RCM signal."

**L396/397:** "in low/higher elevations"    "at low/higher elevations"

**L407:** "meteorology"    "output" or "meteorological output"

**L470:** What do you mean by "two months short snow cover duration"? A shortening by two months? **L474:** "lower horizonal spacing"    "higher horizontal spacing"

**Figures and Tables**

**Figure 3:** Why is there no line in the upmost row in column "DC"? Furthermore, are results shown in this figure  computed with bias adjustment and downscaling (or only the former)?

DC has no past observations of its own. They are actually the same from the RAW column. And results here are from bias adjustment only. We can explain this better in the caption.

**Figure 6:** Interestingly, the line for "bias correction" is closer to "AMUNDSEN" than "downscaling" in the rightmost panel. Can you explain this?

This is the case for RCP8.5. For RCP2.6 this only holds for > 2500m, and below downscaling is closer to AMUNDSEN. Unfortunately, too many factors differ that we cannot test given the methodological setup to answer this question.

**Table 1:** I'm only partially convinced about the point "+ Consistent climate change signal of surface meteorology with snow cover". This is true for the raw RCM output, but no longer for the bias corrected  product.

True. We shall remove this point from the table.

**Figure S1:** Replace "snow plausible" in caption with e.g. "snow accumulation"

Fine.

**Figure S2:** "surrounding" is a bit misleading in the caption – maybe better "aggregated"

Yes, aggregated would work. Or encompassing.

**Figure S11:** caption line 2: "and downscaling."

Yes.

**Figure S12:** I guess the part in the squared brackets belongs to the figure caption?

Yes, it was copied from Fig. 6, but we can put it to the rest of the caption of Fig S12.

**Table S5:** I'm not sure if the reference "See also Figure S6" is correct/intended.

Thanks for spotting this typo. We meant Fig S7.

---

## Author Response (AR1)

**Cover Letter**

Dear Hongkai Gao,

here we present the revised manuscript. We are sorry for the incurred delays and thank you again for the two deadline extensions. This allowed us to work in-depth on the manuscript based on the reviewers comments and related issues. Here a summary of the main changes:

- Consistently used hydrological years throughout the manuscript (no significant differences to before, since all based on 20+ years climatologies, but more consistent now)
- Added an overview figure on the methodology, and rewrote large parts of the methods to make them easier to follow
- Clarified absolute/relative changes in snow cover fraction (percentage points vs. percent) by showing both in most cases
- Added a section on downscaling validation
- Added a section on scale issues in complex topography (which hinders direct comparisons between high and low resolution estimates; only allows comparisons of changes)
- Added a summary table in the conclusion
- Based on a reviewers comment we changed the title to "Bias adjustment and downscaling of snow cover fraction projections from regional climate models using remote sensing for the European Alps" (modified: bias correction → bias adjustment)

Besides, we provide point-by-point replies to the reviewers.

We look forward to the re-evaluation and are available for any further questions or comments,

Michael Matiu, also on behalf of the co-author, Florian Hanzer

**Reply to Referee #1**

This paper used a bias correction and downscaling method to estimate snow cover fraction from regional climate models. The method and results are difficult to follow. Comments and questions are below.

Thank you for the review and constructive feedback.

As the other referee has also pointed out that the methods and results are difficult to follow, we worked on readability in general. We also provided an additional figure to make the procedure(s) easier to understand.

Figure 1, the outline of the small study area is not clear. Please also give the description of the small study area. In Figure c and d, which data source does the snow cover duration come from?

The snow cover duration is from MODIS, based on the same data used in the downscaling procedure. We clarified this in the caption. Also we made the small study area better visible in the map, and provided a small description in the main text.

L95-100 belong to "Study Area", rather than "Data".

We added a separate section titled "Study Area".

Section 2.1 "Observed snow cover fraction from remote sensing", in addition to pointing to referenced paper (Matiu et al., 20201), state how you obtained the daily cloud-free snow cover maps?

The cloud-free maps were obtained by a series of temporal and spatial filters.

- We applied a mean filter over a square of size 299 pixels centered on each pixel, where a snow pixel was reclassified as clouds, if the neighborhood contained more cloud than snow pixels, and a cloud pixel was reclassified as snow, if the neighbourhood contained more snow than cloud pixels. This was performed to improve accuracy at cloud borders or within clouds, where misclassifications sometimes occurred. This step was only performed for April to October, where this issue was mostly prevalent.
- Then, we applied a conservative temporal filter, which is based on the persistence of snow. A cloud pixel was reclassified as snow, if it was snow one or two days before, as well as snow one or two days after. The same for land pixels.
- Afterwards, a spatial filter was applied based on snow and land lines, which exploits the elevational dependence of snow cover. For any given day, all cloud pixels above the average elevation of all snow pixels were reclassified snow, and all cloud pixels below the average elevation of all land pixels were reclassifed as land. This step was not applied if the image had more than 50% clouds (-> snow and land lines not representative), the ratio of snow to land pixels was less than 0.05 (-> too little snow for snow line to be representative), and not in the summer months June to September (-> too little snow)
- Finally, a greedy temporal filter was applied that replaced any cloud pixel with the next available observation (land or snow) from either forward or backward in time. Most values were filled after a few days, but we set a maximum of ten days as the threshold to look for cloudfree observations.

We provided a summary of these steps in the manuscript.

> Change "2.1 Snow cover fraction from regional climate models" to "2.2 Snow cover fraction from regional climate models".

Thanks for spotting this error.

> L155-157, please provide more detailed information for the conversion from snow water equivalent to snow cover fraction. Snow cover fraction is a ratio, how did you use the threshold of 5 mm to get the snow cover fraction.

We used the 5mm threshold to create a binary value for each pixel/grid cell and each day. This binary value was then either snow or snow-free. Snow cover fraction was then calculated either along space, time, or spacetime by calculating the ratio of pixels with snow divided by all pixels along space and time. The manuscript now reads:

> The modelled snow water equivalents were converted into binary snow/land indicators using a threshold of 5 mm. We also evaluated a threshold of 15 mm but found differences to be negligible. Snow cover fraction was then calculated by averaging over time (e.g., months), space (which includes elevation bands), or both.

> L162:why use 19 GCM-RCM for RCP8.5 and 3 GCM-RCM for RCP2.6?

This imbalance between RCP8.5 and RCP2.6 scenarios results from the fact that more RCP8.5 simulations are available than for any other scenarios. We also tried to take only the common GCM-RCM pairs, but found that results for the mean were similar, and more GCM-RCMs give a better estimate of the modelling spread. So, we decided to keep the maximum available number.

We elaborated more on this issue in the manuscript and also made the hint to the little number of scenarios for RCP2.6 more prominent.

> L169-170, the full name of DC, QM, QDM have provided in the Introduction section, here just use DC, QM, QDM.

Done.

> How did you validate the estimated snow cover fraction?

We intended to use the comparison between the physical-based AMUNDSEN model and the statistical-based downscaling as a kind of two-way validation for both approaches. Since these are projected changes for the future, we have no real validation data set, but can only perform comparison betweens approaches and models. For example, the comparison between raw, bias adjusted, and downscaled estimates from RCMs as well as AMUNDSEN can give some hints on the differences and agreements. But without a true validation data set, there is little else possible.

Otherwise, the general capability of RCMs to reproduce snow cover has been validated for the past. We rewrote the evaluation paragraph in the introduction:

> In high mountain regions, evaluations of snow from RCMs are challenging because of a general lack of suitable reference data and scale mismatches between observations and

models. The arguably most relevant snow parameter, snow water equivalent (SWE), is also the most difficult to estimate. In-situ observations are sparse, and estimates based on remote sensing suffer from large uncertainties (Largeron et al., 2020). For the European Alps, Terzago et al. (2017) evaluated SWE from EURO-CORDEX RCMs using an array of remote sensing and reanalysis products, and found a large spread in reference data sets, locally large overestimation of SWE, and differences between GCM and renalysis driven RCMs. Using an interpolated SWE data set based on in-situ data in Switzerland, Steger et al. (2013) found a general underestimation of SWE for elevations below 1000 m and overestimation above 1500 m. On the other hand, Matiu et al. (2020b) focused on different snow parameters, namely snow depth from in-situ observations and snow cover fraction from remote sensing, and found a good agreement between RCMs and observations, when accounting for elevation and temperature differences between observations and models. It is likely that scale mismatches (low vs. high resolution grids or point vs. grid cell), associated elevation biases, and the different reference data set uncertainties are causing these contradicting results.

Finally, the downscaling procedure can be validated internally (as has also been suggested by the other referee). For this, the upscaled MODIS snow cover fraction has been downscaled again and compared to the original maps. We added results to this validation, with some more analysis in a new section 4.4 in the revised manuscript.

**Reply to Referee #2**

**Summary and general comments**

In this study, a statistical bias adjustment and downscaling method for snow cover fraction data is explored. Simulated snow cover fractions are obtained from the EURO-CORDEX regional climate model ensemble at a horizontal resolution of 0.11° – both for the present day climate and for the end of the 21st century (for RCP 2.6 and 8.5). The bias adjustment is performed with MODIS remote sensing data, which is spatially aggregated from a resolution of 250 m to a similar scale as the EURO-CORDEX data. For the bias adjustment, four different methods are initially applied – delta change, quantile mapping, quantile delta mapping (QDM) and multivariate QDM. Bias-adjusted snow cover fractions are then subsequently downscaled to the initial high resolution of 250 m. The entire procedure is, for a subdomain, compared to a rather conventional approach in which snow cover fraction is derived from a high-resolution snow model, which is forced by downscaled RCM output.

This study explores an interesting and novel approach – namely the application of bias adjustment for snow cover fraction. The authors evaluate the suitability of different bias adjustment methods for this challenging approach. The manuscript is predominantly well written – some sections, like the explanation of the downscaling method – are however difficult to follow and should be improved. At some places, a more extensive discussion of the results would be useful – for instance regarding projected relative changes in snow cover fractions – which are surprisingly largest at high elevations.

Thank you for the positive appreciation of our work and your extensive review, which greatly improved the revised version of the manuscript.

Regarding your major points, which you also detailed further below:
- We added an explanatory figure in the methods section and improved the writing in general
- We discuss results more intensively with respect to relative changes by elevation and highlight more the difference between absolute percentage changes in snow cover fraction and relative changes.

**Major comments**

**Remapping of high-resolution MODIS and CORDEX data**
I'm a bit confused about the performed remapping. You first aggregate MODIS data to a low resolution grid and then remap CORDEX data to this grid with a nearest neighbour method, right? This seems to be unnecessarily complicated. Wouldn't it be easier to remap MODIS data (for instance with a conservative method) directly to the default rotated latitude/longitude CORDEX grid?

Yes, your proposition seems at first easier. However, it has been complicated by computational constraints. The support for the rotated CORDEX grid is very low in the software we used, namely R statistical software. At the start of the project (2-3 years ago), it was complicated if not impossible to deal with the rotated grid in R appropriately. Nowadays, support is a little better, but still not good. In the end we used the CDO tools, together with the more standard LAEA grid of MODIS as backbone, and reprojected RCMs.

**Downscaling method**
The downscaling method is very interesting but difficult to follow in some places:
- A general question (just out of curiosity): downscaling can also be performed directly within the bias adjustment method. However, it seems to be less appropriate if observational data is available on a much higher spatial resolution. Did you nonetheless consider this option?

Yes, our initial idea was also to directly apply QM or QDM with observational data. In our case, however, we had to apply it to different types of variables: At 0.11° RCM scale we have a continuous but bounded snow cover fraction (0-100%) and at 250m observation we have a binary snow/land variable. So, we could not apply QM and QDM directly. On a side note: While the MODIS product we used is a binary one that has been developed at the institute, there exist also other options. Eg. the MODIS V6 product from NASA offers the NDSI (normalized difference snow index), from which snow cover fraction could be calculated. So with other observational data, the bias adjustment could be combined with downscaling directly. This would be an interesting alternative, but beyond the scope of this study...

- Line 229: Is the requirement for the function not rather "strictly increasing" to get a unique solution for SNCp50? Furthermore, I don't understand the subsequent part with the "longest non-strictly increasing subsequence"

Actually, non-strictly increasing suffices. SNCp50 occurs when the y=0.5 line is crossed (e.g. Figure 2 in the manuscript, or figure below). For this, it is not important if we have a flat (non-increasing) portion (topright in figure below). The only problem with non-strictly arises if the non-increasing flat portion is at exactly y=0.5; however, this situation is extremely rare (if it happened at all).

What destroy the uniqueness is when the y=0.5 is crossed twice or more times (bottomright of figure below). While even not monotonic sequences can have a unique solution (bottomleft of figure below), by having a monotonic sequence, uniqueness is guaranteed.

[Figure]

Regarding the second point, if the curve is decreasing at some point or even at multiple points, then there are multiple choices to select a monotonic subsequence by removing some points. So we selected the longest possible of these, assuming it is the best estimate. The solution to selecting the longest subsequence is not unique, however, the length of the monotonic subsequence has a maximum. See also this article on wikipedia for more info: https://en.wikipedia.org/wiki/Longest_increasing_subsequence

The new Figure 2 in the manuscript (method overview) also shows some examples of the selection of the monotonic subsequence.

- I found it particularly difficult to follow the "filling of missing SNCp50 values" section (lines 235 – 248). The comprehensibility might be improved with an additional sketch or figure. Moreover, I have some specific questions:
    - Could you explain why the Wasserstein distance is a suitable metric for this problem?  o What do you mean by "no one-to-one correspondence"?
    - By nearest low-resolution pixels (line 240), you refer to the Wasserstein distance, not the horizontal distance, right?
    - Line 247: how do you determine these 100 pixels exactly from the theoretically 50 * 100 available pixels?
    - How did you proceed with the very small fraction (<0.001%) of pixels with still missing SNCp50 values?

That is a good idea. We provided an additional figure in the revised manuscript that explains the whole set of methods (from bias adjustment to downscaling). We also revised the text in general, and also included your specific points:

Regarding your specific points:
- The Wasserstein distance works for two sets of values that do not have to have a one-to-one correspondence between observations, such as, for example, probability distributions. The elevations of high-res pixels can be considered such a "probability distribution". If we used other metrics, such as Euclidean distance, then we need to have one-to-one correspondence between the high-res elevations of two low-res grid cells. Such a correspondence could be e.g. by always comparing the topleft high-res pixel of low-res grid cell A with the topleft high-res pixel of low-res grid cell B, … , and the bottomright to the bottomright. But this would create not useful distances for the task, which is to compare the sub-grid elevation distributions.
- One-to-one correspondence: see previous comment.
- Exactly, at line 240 we refer to the Wasserstein distance.
- At line 247, we order the pixels by elevation difference.
- The rest of the <0.001% pixels were removed from the analysis.

- Line 249: Could the downscaling approach not simply be evaluated by reconstructing high-resolution MODIS snow cover from the spatially aggregated pixels?

That is true. We applied it directly to the upscaled observed MODIS data and added another section on the validation of the downscaling approach (new Section 4.4).

- It seems that you assume a seasonally stationary downscaling relation, i.e. there is no temporal dependency. Is this assumption valid? Or could the relation look slightly different for e.g. late autumn and spring?

Yes, we assume a seasonally stationary downscaling relation. The main reason is that the downscaling is based on terrain morphology, which stays the same over the season.
The main components of the change in snow cover fraction are snow deposition driven by snowfall and snow disappearing from snow melt. The strongest element in both snowfall and melt is elevation - at least at the scales used here in the 250m range. And this link to elevation is what the downscaling procedure extracts (and also north south exposition, and other local features).
Once we go up to higher resolutions, such as 10-20m, where preferential deposition of snow, terrain shading, and wind start to play strong roles - then the seasonal stationarity is questionable, since it will strongly depend on whether it's the start of the season or end of the season.

We tested the seasonal stationarity assumption by deriving two different SNCp50 values, one for the first half-year (start of season, Sep - Feb) and one for the second (end of season, Mar-Aug). But we did not find large systematic differences. We added a paragraph on this issue in the validation section 4.4.

**Projected changes in snow cover fraction for different elevations**
I'm surprised that the elevation gradients in Figure S12 are so distinctively different. Could this also be caused by the different RCM ensembles used for AMUNDSEN and the statistical approach (i.e. do different RCMs for instance show different elevation gradients in warming)? It is generally a bit counter-intuitive that the largest relative reduction in snow cover fraction is projected for the highest elevations (see Figure S10b and S12).

Yes, we too were surprised about the large differences in the elevation gradients in old Figure S12. While beyond the scope of the study, it would be very interesting to understand where these differences come from. However, this would require a different experiment setup (same RCM-GCMs, with and without bias adjustment, …).
Regarding your questions: Yes, different RCMs show different elevation gradients, see e.g. revised manuscript Fig S8 or S11. But while the model ensemble affects the spread, for the mean signal, the difference between the ensembles used for AMUNDSEN and the statistical approach should be little.

In old Figures S10b and S12, we show the difference in percentage points, which is the absolute change in snow cover fraction, whose units are already percent. These are not relative changes. We added figures with true relative changes for both bias adjustment and downscaling (new Figures S9, S11, also in the main manuscript Figure 6), which show the expected behaviour: largest relative reductions at low elevations. However, for the Ötztal Alps subregion, DS still shows the counter-intuitive behaviour (but not for the whole study region, ie the whole Alps). We discuss this in the manuscript as follows:

This elevation gradient in the relative SNC changes from downscaling for the Ötztal Alps is counter-intuitive. It is also different from the gradients for bias adjustment and dowscaling for the whole study area, which themselves are similar to the results from AMUNDSEN for the Ötztal Alps (Figure 6). One reason for this discrepancy might be that the Ötztal Alps region comprises only 15 RCM grid cells with a very limited elevation range (1800 to 2800 m), which has to be extrapolated to a much wider elevation range (900 to 3700 m) in the finer spatial resolution. In the case of AMUNDSEN, this extrapolation is performed on the surface meteorology, which seems to work better than the extrapolation performed in the SNC downscaling approach.

Additionally, when looking deeper into this, we found scale issues when comparing high and low resolution data sets (irrespective of bias adjustment and downscaling; already for MODIS SCD). These hinder direct comparisons of elevation gradients. We added a separate section on these "scale issues" in the methods. It still allows comparisons of change estimates by elevation but not of actual SCD/SNC elevation gradients between two different spatial resolutions.

**Minor comments**

**Content-related (text)**
**Line 15:** With which bias adjustment method were these number computed?

From QDM, we added it to the abstract.

**L17:** "The comparison of the statistical" → This evaluation confirms the robustness of the downscaling method → I would therefore mention it before presenting results for the future climate.

Makes sense. We restructured the abstract, showing first bias adjustment results, followed by downscaling, and not mixing the two.

**L19:** The term "bias correction" is a bit outdated. Better use the term "bias adjustment" (see e.g. https://cordex.org/data-access/bias-adjusted-rcm-data/ and https://hypeweb.smhi.se/what-is-bias-adjustment/)

Thank you for pointing this out. We changed the terminology throughout the manuscript and also in the title.

**L19:** "plausible" might be a bit vague. Maybe better "more suitable" or "applicable".

We changed it to "more suitable".

**L37:** I find it difficult to follow this sentence. Maybe it's better to write "because climate change violates the assumption of stationarity" (if you mean that).

That's what we meant. We used your suggestion.

**L53:** "…and even snow depth" → this part seems to be a bit contradictory to the previous sentence. Snow depth is reproduced relatively well but SWE not?

We rewrote the whole paragraph on evaluation of snow from RCMs to better reflect these issues and contradictions. This also takes into account your next comment. The manuscript now reads:

> Additionally, RCMs suffer from biases, for instance in temperature and precipitation (Vautard et al., 2021), which would be the meteorological forcing for dedicated snow models, but also from biases caused by the relatively simple snow schemes of RCMs. In high mountain regions, evaluations of snow from RCMs are challenging because of a general lack of suitable reference data and scale mismatches between observations and models. The arguably most relevant snow parameter, snow water equivalent (SWE), is also the most difficult to estimate. In-situ observations are sparse, and estimates based on remote sensing suffer from large uncertainties (Largeron et al., 2020). For the European Alps, Terzago et al. (2017) evaluated SWE from EURO-CORDEX RCMs using an array of remote sensing and reanalysis products, and found a large spread in reference data sets, locally large overestimation of SWE, and differences between GCM and renalysis driven RCMs. Using an interpolated SWE data set based on in-situ data in Switzerland, Steger et al. (2013) found a general underestimation of SWE for elevations below 1000 m and overestimation above 1500 m. On the other hand, Matiu et al. (2020b) focused on different snow parameters, namely snow depth from in-situ observations and snow cover fraction from remote sensing, and found a good agreement between RCMs and observations, when accounting for elevation and temperature differences between observations and models. It is likely that scale mismatches (low vs. high resolution grids or point vs. grid cell), associated elevation biases, and the different reference data set uncertainties are causing these contradicting results.

**L54:** "gridded estimates of SWE" → You refer to data derived from in-situ measurements (and not models), right? If so, I would explicitly state this. Furthermore, I guess you want to allude to the spatial representativeness of different data sets (in-situ vs. remote sensing) here. If so, you could discuss this in an additional sentence here…

See answer to comment above.

**L61:** "The reference observations can be points or grids, are often limited in extent compared to RCMs, and feature, in case of grids, typically higher resolutions."

Thank you for the proposition. We changed it.

**L82:** "The motivation is that…" → this sentence is rather long and difficult to read – could you rephrase it? I furthermore do not understand the end "…while future change estimates should be consistent." Consistent with what?

Consistent across time. So, if we assume systematic biases, these should be consistent across time. We rephrased to:

The motivation behind the statistical adjustment of snow cover fraction from RCMs is that the biases are systematic. They were shown to be mainly caused by orography and temperature, partly also precipitation, mismatches (Matiu et al., 2020b). These systematic biases are consistent across time, and thus future change estimates can be statistically adjusted.

**L84:** "While bias correction…" → I don't understand this sentence entirely. What is the link between absolute values and bias adjustment?

If you have biased values, e.g., temperature, then your climate change information (e.g., number of tropical nights) or impact model is also biased and not representative. So unbiased absolute values are better in this regard. We rephrased and shortened the sentence to:

While bias adjustment cannot add information beyond what is contained in the RCM, it can reduce model spread. Additionally, it can make information on future projections more meaningful compared to solely providing change estimates, which are sometimes hard to interpret. Unbiased absolute values are better for climate change information and for impact assessments, which often depend on absolute thresholds, and for which biased estimates would not be representative.

**L118:** Did you apply a nearly or entirely cloud-free data set in the end?

It was a nearly cloud-free data set. Corrected in the manuscript.

**L119:** What does "nominal" and "effective" resolution mean in this context?

Actual map resolution is 232 m, but we used 250 m (approximation) throughout the manuscript for simplicity. We added a sentence to clarify it.

**L127:** "all variables" → which variables do you mean?

While the coverage for temperature and precipitation is high, snow variables such as snow cover fraction are not available for all models. We added a sentence in the manuscript to clarify this.

**L139:** I was wondering – would it also be possible to account for the snow accumulation issue within the bias adjustment method?

We think not. If we assume a fully snow covered grid cell across the whole simulation period, this would likely render the change estimates for this grid cell inadequate. For SWE, the change would be too positive, for snow cover fraction, no change would occur. The bias adjustment might pull the hindcast closer to observations, but it cannot "adjust" for this accumulation for the future. The trend-preserving approaches will then show increasing trends, caused by snow accumulation.

**L160:** What do you mean by "single GCM-RCM biases"?

Sorry for the confusion. What we meant was that biases for the calibration period were removed. Actually, this was more the problem with different baselines for calibration, since for AMUNDSEN 1970-2005 was the calibration period, while here it's 2001 to 2020. The manuscript now reads:

In addition, the QM in Hanzer et al. (2018) has been applied using the period 1970 to 2005 as baseline, while the baseline in this study was 2001 to 2020.

**L171:** I would be careful that the reader is not confusing calendar with hydrological years. I guess you conduct all your evaluations for hydrological years – right? Then I would state the definition of a hydrological year somewhere and mention, that all years refer to hydrological years – unless otherwise stated.

Thanks for pointing out this inconsistency. Actually, we used both calendar and hydrological years initially. However, for the revised manuscript we switched to hydrological years consistently for all data sets and methods. We followed your suggestion and now the manuscript reads the following at the end of Sec. 2.2.:

To derive annual snow cover duration (SCD) maps, we used hydrological years defined such as to maximize the available data within the MODIS period. The split was in summer, which is anyway the least important period for seasonal snow in mountains. A hydrological year is defined here as starting August 1 and ending July 31, and designated by the year it ends. The past SCD climatology (Fig. 1) is thus based on the (hydrological) years 2001 to 2020, which covers the period 2000-08-01 to 2020-07-31. For simplicity, the

term year will be used as substitute for hydrological year from now on, thus also when referring to the climate model data.

**L175:** I would remove ", usually 1970 – 2000"

Done.

**L189:** "also to the maximum of 1." → this is oddly phrased. I guess values above 0.999 are set to 1.0, right?

Exactly. We rephrased to:

The trace condition, which sets all values below a threshold (here: 0.001, also called trace value) to exact zeros, has also been applied to the maximum, so that all values above 0.999 were set to exact ones.

**L188:** The trace condition was applied to the about of the Q(D)M algorithm - right?

Yes. We added a hint to this at the beginning of the paragraph.

**L195:** "breaks the temporal consistency in the bias corrected SNC time series" → what do you mean by that?

The random component is added in a distributional manner for each month and 30 years of data. So for climatological averages for that month, that should be fine. But at the daily scale, this random component can cause, e.g., that even during the melting period snow cover fraction can increase. And also the year-to-year variability might not be representative. We added the following in the manscript:

Since it's applied in a distributional manner over all days in each month for a 20/30 year period, the monthly climatologies are fine. But at the daily scale, the random component might lead to inconsistencies, such as sudden jumps in the snow cover fraction time series or increasing snow cover fraction in the melt season. Consequently, no estimates of interannual variability can be calculated.

**L260:** What do you mean by "preferred"? Are trends only preserved for ratio variables?

No, for QDM, trends are preserved also for the interval variables. What we meant here was that, the trend-preserving attribute is preferred for ratio variables in general. In addition, QM suffers from trend modification, and DC only adjusts the mean. We rewrote the sentence to:

The downscaling was applied for QDM bias adjusted low-resolution snow cover fraction only, and not for the other bias adjustment methods. QM showed artificial modification of trends (see Sec. 4.1), and DC is theoretically inferior to QDM, since DC only adjusts the mean, while QDM adjusts the whole distribution.

**L388:** What do you mean by "analogue based"?

We meant climate analogues, which is a downscaling technique that looks for similar (analogue) observations in the past. We rephrased the sentence and added references:

For the future, other downscaling techniques could be explored, such as analogue, perfect prog, or weather typing methods (Zorita and Storch, 1999; Gutiérrez et al., 2013), as well as, spatially explicit gridded downscaling approaches (Werner and Cannon, 2016).

**Typos, phrasing and stylistic comments**
**General:** Both the terms grid cell and pixel are used. To avoid ambiguity, it would be good to use them consistently: You could for instance use grid cell exclusively for climate models and pixel for remote sensing data.

Good idea. We consistently use the terming you suggested. We also added an explanatory sentence:

From now on, the term pixel shall refer to the high-resolution area (250 m by 250 m) and grid cell to the coarse-resolution area (0.11° by 0.11°), for both MODIS and RCMs.

All suggestions below were adopted as they are, unless otherwise stated.

**L22:** remove "only"

**L28:** "consequences in water supplies" sound odd. Maybe rather "on"?

**L31:** maybe better "snow cover causes a significant atmospheric feedback due to its high albedo" **L50:** "such as for" sounds a bit odd. Maybe better: "for instance in"

**L51:** "but also from biases in RCM snow cover." I would rephrase this to something like: "but also from biases caused by the relatively simple snow schemes of RCMs."

**L67:** I would rearrange this sentence to: "Since QM has been show to modify trends in a few cases, quantile delta mapping (QDM) was developed, which represents a trend-preserving QM approach."

**L77:** "We restrict the study to snow cover fraction, which is, in contrast to snow depth and SWE, globally available on a high spatial resolution. The presented method has therefore a global potential for application."
Thank you for the proposition. We just added "and with high accuracy" after spatial resolution, since global SWE and snow depth products exist, but their accuracy is limited or questionable, in addition to spatial resolution constraints.

**L82:** "meteorology" → "output"

**L88:** I would replace "representation" by "pattern". "Representation" can be ambiguous in this context (one could understand that snow cover representation in RCMs is improved).

**L91:** "is as" → "is structure as"

**L95:** The study region (Fig. 1) encompasses the European Alps and spans from ~43 – 48.5° N and ~5 – 17° E, which roughly corresponds to the Greater Alpine Region (Auer et al., 2007).
Proposition adopted, with slight adjustments for specifying the long-lat extent.

**L133:** "although snow depth match very good" → "although snow depths are well reproduced"

**L172:** "merged to" → "merged with"

**L270:** "under- and overestimating" → "simulated"
We feel your proposition is not what we meant. We changed the whole sentence to:
> Additionally, the monthly areal averages of snow cover fraction were over- and underestimated, depending on both RCM and GCM (Fig. 3), as has also been shown previously

**L305:** change "lower ensemble" to "smaller ensemble"

**L325:** change "a more of a fully snow free and fully snow-covered grid cell over time" to "more often fully snow free or snow-covered conditions over time"

**L363:** I would not write "became" if you compare two hypothetical projections
Changed to "was".

**L383:** Change to: "In contrast, the final imputation step of SNCp50 is based on a simple elevational dependence of snow cover, and could thus directly be estimated from a low-resolution RCM signal."

**L396/397:** "in low/higher elevations" → "at low/higher elevations"

**L407:** "meteorology" → "output" or "meteorological output"
We chose "output".

**L470:** What do you mean by "two months short snow cover duration"? A shortening by two months?
Exactly. We clarified it.

**L474:** "lower horizonal spacing" → "higher horizontal spacing"
If we define spacing as Blöschl 1999, then both, your and our, wording can be misleading. Lower can imply less space between grid cell centers (a finer grid) or lower resolution (a coarser grid), and vice versa. We used "finer" instead, which is hopefully less ambiguous.

**Figures and Tables**

**Figure 3:** Why is there no line in the upmost row in column "DC"? Furthermore, are results shown in this figure computed with bias adjustment and downscaling (or only the former)?

DC has no past observations of its own. They are actually the same from the RAW column. And results here are from bias adjustment only. We modified the caption to cover both questions.

**Figure 6:** Interestingly, the line for "bias correction" is closer to "AMUNDSEN" than "downscaling" in the rightmost panel. Can you explain this?

One issue in old Figure 6 is concerning the scale issues, which introduces biases between high and low resolutions (thus also between bias adjustment and downscaling); see also new Section 2.2.1. Now in the revised manuscript Figure 6 only shows change estimates, where bias adjustment and downscaling are similar, but differ from

AMUNDSEN. At least within the Ötztal Alps. But AMUNDSEN gradients match whole study area gradients from bias adjustment and downscaling quite well.

**Table 1:** I'm only partially convinced about the point "+ Consistent climate change signal of surface meteorology with snow cover". This is true for the raw RCM output, but no longer for the bias corrected product.

True. We removed this point from the table and added it to the text, since it applies to both approaches.

Instead we added another point in the table: Dedicated snow models require surface meterology, which often involves in-situ data combined with some form of downscaling of RCM output (-), while observational data for the bias adjustment/downscaling of snow cover fraction is readily available from remote sensing (+).

**Figure S1:** Replace "snow plausible" in caption with e.g. "snow accumulation"

Done.

**Figure S2:** "surrounding" is a bit misleading in the caption – maybe better "aggregated"

We changed to "encompassing".

 **Figure S11:** caption line 2: "and downscaling."

Done.

**Figure S12:** I guess the part in the squared brackets belongs to the figure caption?

Yes, it was copied from Fig. 6, but we can put it to the rest of the caption of Fig S12. Note that figure numbers might have changed during the revision.

**Table S5:** I'm not sure if the reference "See also Figure S6" is correct/intended.

Thanks for spotting this typo. We meant Fig S7. Note that figure numbers might have changed during the revision.

---

## Referee Report (RR1)

It seems the author have put a lot of effort in the revised version of the manuscript – this is very appreciated. I think particularly the methodology section was much improved, which enhances the comprehensibility and potential reproducibility of the novel approaches described in the manuscript. Most of my comments are of minor but there is one larger issue that should be addressed/resolved. The page and line number refer to manuscript version 2.

**Major comment**

I'm puzzled by the sometimes very different elevation gradients in projected (absolute) snow cover change, which result from raw RCM data, bias adjustment and downscaling. E.g. in Figure S8, you show that for RCP8.5, largest absolute reduction in snow cover fractions are expected for elevations ~1000 – 2000 m, depending on the RCM-GCM combination, season and processing method (raw or different bias adjustments). In contrast, Figure 6 (rightmost panels) shows that absolute reduction in snow cover fraction generally increase with elevation. Both figures show bias adjusted data (without downscaling) – so I guess one of them must be wrong.

**Minor comments**

**L91:** I'm not sure if really all biases are constant in time. I suggest to rephrase this sentence to: " these systematic biases seem to be predominantly constant across time, …"
**L98:** "limitations of both approaches." → limitations of the high-resolution snow model are not discussed, I would thus rephrase it to "limitations of the presented method."
**L161:** I'm not sure if I understand this sentence correctly – with "future change estimates" you mean relative values, right?
**L187:** I would rephrase (or entirely remove) the sentence with "unavoidable" because you just propose in the following sentence how the imbalance could be resolved.
**L196:** "future meteorology" → "projected meteorological data"
**L281:** I'm confused by the term "non-strictly" because a "strictly monotonic relationship" would also guarantee a unique solution (as you show in the reply to referee 2). Maybe the term "monotonically non-decreasing function" would be less ambiguous in this context?
**L294-L298:** I was not able to fully understand what you mean by these lines.
**L472:** The ordering of the subsections 4.x is a bit odd. Normally, the evaluation/validation of the results is shown first before one discusses e.g. future projections (to underline the robustness of the results). This ordering is however almost reversed in your manuscript. You could consider reordering the subsections.

**Figure 2:** This figure is very helpful to understand the methodology. Could you increase the axis (tick) labels in panel (b) a bit – they are rather hard to read.
**Figure S2:** I'm not sure what you mean by "SCD summaries by elevation." Maybe you could rephrase that. Besides, I was quite confused by the x-axis labelling of panel (b) "pixels/grid cells". I first read it as "pixels per grid cells" but I think you mean "pixels or grid cells".

---

## Author Response (AR2)

**Cover Letter**

Dear Hongkai Gao,

here we present the revised manuscript. We addressed all further reviewer comments and provide below the point-by-point replies.

Regarding the file validation remark on using colorblind friendly color scales: We generally use color palettes that are perceptually uniform and respect color vision deficiencies, and are possibly even print-safe, such as https://www.fabiocrameri.ch/colourmaps/, https://colorbrewer2.org/, or viridis. The only figure that might not respect this, is Figure 3 (and some similar figures in the supplement). However, in these cases, the different colored lines should only give an impression of variability, and it's not necessarily needed to follow every line to understand the message (which might be challenging for people with vision deficiencies, but partly also for people without deficiencies).

Best regards,

Michael Matiu, also on behalf of the co-author, Florian Hanzer

**Reply to Referee #1**

You made good responses to reviewers' comments. The added descriptions about datasets, validation, and discussions that improved the manuscript.

Comment 1:
Figure 2a, 2b, and 2c is not well organized. It is suggested to provide a overall title for figure 2, and to show the links among figure 2a, 2b, and 2c.

Thank you for the evaluation and the comment. We improved figure 2 as suggested. We added titles to parts a, b, and c, and provide the links between the overall summary (a) and detailed substeps (b) and (c).

**Reply to Referee #2**

It seems the author have put a lot of effort in the revised version of the manuscript – this is very appreciated. I think particularly the methodology section was much improved, which enhances the comprehensibility and potential reproducibility of the novel approaches described in the manuscript. Most of my comments are of minor but there is one larger issue that should be addressed/resolved. The page and line number refer to manuscript version 2.

Thank you for the positive feedback.

**Major comment**

I'm puzzled by the sometimes very different elevation gradients in projected (absolute) snow cover change, which result from raw RCM data, bias adjustment and downscaling. E.g. in Figure S8, you show that for RCP8.5, largest absolute reduction in snow cover fractions are expected for elevations ~1000 – 2000 m, depending on the RCM-GCM combination, season and processing method (raw or different bias adjustments). In contrast, Figure 6 (rightmost panels) shows that absolute reduction in snow cover fraction generally increase with elevation. Both figures show bias adjusted data (without downscaling) – so I guess one of them must be wrong.

The difference in the elevation gradients that you mention is because Figure 6 shows annual values, while Figure S8 shows seasonal values, but not for all seasons (only winter and spring).

We added two additional figures in the supplement (S18 and S19 in the new version). An annual version of Figure S8, which should now resemble better what is seen in Figure 6. And a seasonal variant of Figure 6, which should be then more similar to Figure S8.

Additionally we added "seasonal" or "annual" to all figure legends and figure axes in the main manuscript and supplement. We hope this helps in reducing confusion and assures you that the figures are correct.

**Minor comments**

L91: I'm not sure if really all biases are constant in time. I suggest to rephrase this sentence to: " these systematic biases seem to be predominantly constant across time, …"

Done.

L98: "limitations of both approaches." à limitations of the high-resolution snow model are not discussed, I would thus rephrase it to "limitations of the presented method."

True, corrected.

L161: I'm not sure if I understand this sentence correctly – with "future change estimates" you mean relative values, right?

Yes, relative, but also absolute values. To be clearer, we rephrased to the following:

> Consequently, also for the future maps, SCD cannot be compared between high and low resolutions without introducing the same errors from scale issues. This holds for the absolute number of SCD, i.e., how many days with snow cover are there at a specific location/elevation. However, it's still possible to compare future absolute and relative change estimates, i.e., how many less or more days with snow cover are there. These change estimates should be unbiased, since subtracting past from future values also subtracts the biases introduced by scale mismatches.

L187: I would rephrase (or entirely remove) the sentence with "unavoidable" because you just propose in the following sentence how the imbalance could be resolved.

Done.

L196: "future meteorology" à "projected meteorological data"

Done.

L281: I'm confused by the term "non-strictly" because a "strictly monotonic relationship" would also guarantee a unique solution (as you show in the reply to referee 2). Maybe the term "monotonically non-decreasing function" would be less ambiguous in this context?

In mathematical terms, non-strictly monotonic encompasses also strictly monotonic (but not vice-versa), so non-strictly is a less strong assumption than strictly, which also includes it. But since it's too confusing, we removed "non-strictly" and only talk about monotonic relationships.

L294-L298: I was not able to fully understand what you mean by these lines.

We are sorry for not being clear. Our intent with the sentences was to motivate the Wasserstein distance. The lines now read:

> We expect two grid cells to be similar, if the two distributions of pixel elevations within the respective grid cells are similar. The Wasserstein distance is especially designed for comparing distributions: If the two distributions are thought of as earth piles, it calculates how much and how far "earth" has to be moved, such that the two distributions agree. Other distance metrics, such as Euclidean, would in this case require to have a pairing of all values (pixel elevations) between the two grid cells, and are not well suited to compare distributions.

L472: The ordering of the subsections 4.x is a bit odd. Normally, the evaluation/validation of the results is shown first before one discusses e.g. future projections (to underline the robustness of the results). This ordering is however almost reversed in your manuscript. You could consider reordering the subsections.

Thank you for the suggestion. We moved the validation of the downscaling from 4.4 to 4.2 after bias adjustment, but before the results of downscaling.

Figure 2: This figure is very helpful to understand the methodology. Could you increase the axis (tick) labels in panel (b) a bit – they are rather hard to read.

Done.

Figure S2: I'm not sure what you mean by "SCD summaries by elevation." Maybe you could rephrase that. Besides, I was quite confused by the x-axis labelling of panel (b) "pixels/grid cells". I first read it as "pixels per grid cells" but I think you mean "pixels or grid cells".

By "SCD summaries" we meant SCD averaged by elevation. We modified the legend accordingly. Thanks for pointing out the ambiguity of panel (b). We modified the x-axis label to "pixels or grid cells".